# Aurora A-dependent CENP-A phosphorylation at inner centromeres protects bioriented chromosomes against cohesion fatigue

Grégory Eot-Houllier[1], Laura Magnaghi-Jaulin[1], Géraldine Fulcrand[1], François-Xavier Moyroud[1], Solange Monier[1] & Christian Jaulin[1]

Sustained spindle tension applied to sister centromeres during mitosis eventually leads to uncoordinated loss of sister chromatid cohesion, a phenomenon known as "cohesion fatigue." We report that Aurora A-dependent phosphorylation of serine 7 of the centromere histone variant CENP-A (p-CENP-AS7) protects bioriented chromosomes against cohesion fatigue. Expression of a non-phosphorylatable version of CENP-A (CENP-AS7A) weakens sister chromatid cohesion only when sister centromeres are under tension, providing the first evidence of a regulated mechanism involved in protection against passive cohesion loss. Consistent with this observation, p-CENP-AS7 is detected at the inner centromere where it forms a discrete domain. The depletion or inhibition of Aurora A phenocopies the expression of CENP-AS7A and we show that Aurora A is recruited to centromeres in a Bub1-dependent manner. We propose that Aurora A-dependent phosphorylation of CENP-A at the inner centromere protects chromosomes against tension-induced cohesion fatigue until the last kinetochore is attached to spindle microtubules.

[1] Institut de Génétique et Développement de Rennes, Epigenetics and Cancer group, Université Rennes 1, UMR 6290 CNRS, 35043 Rennes cedex, France. These authors contributed equally: Grégory Eot-Houllier, Laura Magnaghi-Jaulin. Correspondence and requests for materials should be addressed to G.E.-H. (email: gregory.eot@univ-rennes1.fr) or to C.J. (email: christian.jaulin@univ-rennes1.fr)

The centromere is a specialized domain of the chromosome required for the faithful segregation of sister chromatids during mitosis. In higher eukaryotes, centromere identity is dependent on the incorporation of specific nucleosomes containing CENP-A, a centromere-specific variant of histone H3[1]. However, centromere chromatin does not consist exclusively of CENP-A nucleosomes[2–4], as canonical histone H3 nucleosomes are also present at the centromere. In all models of centromere chromatin structure in mitotic chromosomes, the CENP-A nucleosomes are found exclusively on the external face of the centromere, whereas the inner centromere contains exclusively H3 nucleosomes[2, 4–7]. The N-terminal tail of CENP-A differs considerably from that of H3, and between species. This domain has been reported to be necessary for recruiting kinetochore components and for accurate chromosome segregation[8, 9]. The N-terminal domain of histones is subject to numerous post-translational modifications (PTMs) that influence all chromatin-related processes and PTMs of CENP-A have also recently been described[10–12]. It has been suggested that p-CENP-AS7 is involved in the maintenance of an active kinetochore during mitosis[8], in chromosome alignment and correct chromosome segregation[9], and in the completion of cytokinesis[13]. CENP-AS7 is partially phosphorylated by Aurora A during the prophase of mitosis[9], and this phosphorylation is required for Chromosomal Passenger Complex (CPC) recruitment at centromeres during prometaphase and for the completion of CENP-AS7 phosphorylation by the Aurora B kinase[9, 13–15]. Both the Aurora kinases are involved in various processes required for accurate cell division. Aurora A localizes to spindle poles, where it is involved in centrosome maturation and separation[16]. Aurora B is recruited to the inner side of centromeres, between sister kinetochores, where it supervises chromosome biorientation by correcting erroneous microtubule/kinetochore attachments[17].

One of the major functions of the inner centromere is protecting the attachment between sister chromatids until the spindle assembly checkpoint (SAC) is satisfied. Sister chromatid cohesion must persist at the centromeres during the early stages of mitosis for correct chromosome biorientation and the establishment of tension between sister centromeres. Sister chromatid cohesion is mediated by the cohesin complex[18]. In vertebrates, the phosphorylation of the cohesin complex components by CDK1, PLK1 and Aurora B leads to the removal of most cohesins from the chromosome arms during prophase[19–21]. Until the onset of anaphase, cohesion at centromeres is protected by Shugoshin1 (Sgo1)[22, 23]. Bub1-mediated phosphorylation of the threonine 120 residue of histone H2A (H2AT120) at the centromeres is essential for Sgo1 recruitment during mitosis[24]. However, other mechanisms, including interaction with H3K9Me3-associated heterochromatin protein 1 and direct binding to the cohesin complex, are required for the targeting of Sgo1 to the centromere[23, 25, 26]. Moreover, in human cells, Sgo1 undergoes tension-dependent relocation from the inner centromere to the kinetochores[27, 28]. Sgo1 acts as a sensor of tension between sister kinetochores and promotes chromosome biorientation[29, 30]. However, despite the protective function of Sgo1, sustained tension between sister centromeres ultimately leads to progressive loss of sister chromatid cohesion. This stochastic and unprogrammed phenomenon is known as "cohesion fatigue[31, 32]."

We report here that, unlike the non-phosphorylated form of CENP-A, p-CENP-AS7 localizes to the inner side of the centromere during mitosis. The prevention of CENP-AS7 phosphorylation or the depletion of Aurora A increases the number of cells displaying premature sister chromatid separation (PSCS). We show that, in addition to its known spindle pole localization, Aurora A is associated with centromeres during mitosis. We found that, besides its known role in Sgo1 targeting to

centromeres, the Bub-1 kinase is required for recruiting Aurora A to centromeres and, thus, for CENP-AS7 phosphorylation. The loss of p-CENP-AS7 was found to weaken the binding of Sgo1 to centromeres and to lead to PSCS when the sister centromeres are under tension. Overall, we show that Aurora A-dependent CENP-AS7 phosphorylation is an inner centromere chromatin mark involved in protecting bioriented chromosomes against cohesion fatigue.

## Results

**p-CENP-AS7 localizes to the inner side of centromeres.** Various models have been proposed for the structure of centromere chromatin in mitotic chromosomes. In all these models, CENP-A nucleosomes are found exclusively on the external face of the centromere, whereas the inner centromere contains exclusively H3 nucleosomes. We found that CENP-A localized, as expected, to the outer side of the centromere (Fig. 1a red signal). By contrast, p-CENP-AS7 was detected on the inner side of the centromere, between sister kinetochores (Fig. 1a green signal). The depletion of endogenous CENP-A led to the loss of both forms of CENP-A (Supplementary Fig. 1A), confirming the specificity of the antibody used. This observation was confirmed with two other unrelated anti-p-CENP-AS7 antibodies (Supplementary Fig. 1B). Of the three anti-p-CENP-AS7 antibodies, the rabbit monoclonal antibody against p-CENP-AS7 Millipore 04-792 (used in Fig. 1a and Supplementary Fig. 1A) had the most resistant binding to its target in competition with an unphosphorylated CENP-A peptide or with an unrelated H4 peptide (Supplementary Fig. 1C). We therefore used this antibody in subsequent studies.

Detailed analysis revealed that CENP-A had a narrower distribution than p-CENP-AS7, as shown by its location restricted to the outer side of the centromere (Fig. 1a,b). The distribution of p-CENP-AS7 was more heterogeneous, with this phosphorylated protein detected on the inner centromere in 88% of chromosomes and exclusively on the outer side of the centromere in only 12% of the mitotic cells examined (Fig. 1b). In 54% of the chromosomes, p-CENP-AS7 was found solely at inner centromeres, in various patterns. By contrast, in 34% of chromosomes, p-CENP-AS7 was recruited to both the inner and outer centromere (Fig. 1b). Unlike CENP-A, p-CENP-AS7 frequently presented more diffuse distribution, suggesting that p-CENP-AS7-containing chromatin may be less condensed or more dynamic than its non-phosphorylated counterpart.

We further investigated the structure of p-CENP-AS7-containing chromatin by stretching mitotic centromere chromatin fibers (CFs). In stringent, high-salt conditions, CENP-A-positive mitotic CFs could be extended to 5 to 20 μm (Fig. 1c), whereas CENP-A CFs from interphase cells could be stretched to up to 50 μm (Supplementary Fig. 1D). This difference in stretching capacity between mitotic and interphase centromere chromatin may reflect the highly condensed state of mitotic chromosomes, but is also consistent with the reported cell cycle regulation of centromeric chromatin compaction[33]. On almost all the extended mitotic CFs (14 of 15), CENP-A and p-CENP-AS7 occupied 2 distinct, partially overlapping, domains (Fig. 1c and Supplementary Fig. 1E fibers 2–14). p-CENP-AS7 blocks alternated with CENP-A blocks over a quarter of the length of the fiber, with little or no colocalization. A p-CENP-AS7-positive region next to this alternating pattern excluded any detectable CENP-A blocks. This region spanned the remaining three quarters of the total centromere fiber. This extended centromeric chromatin organization profile is consistent with the patterns observed on chromosome spreads (Fig. 1a,b), in which CENP-A and p-CENP-AS7 occupy non-overlapping regions of the

centromere in most cells. However, in one of the 15 observed stretched mitotic CFs, CENP-A blocks alternated with p-CENP-AS7 blocks over the entire length of the fiber, with the two signals rarely overlapping (Supplementary Fig. 1E, fiber 15). The number of observations was low, due to the technical difficulties involved in obtaining mitotic centromere CFs, but these findings are consistent with the results shown in Fig. 1b, in which CENP-A and p-CENP-AS7 are strictly colocalized on the external centromere in only 12% of metaphase chromosomes.

Our data show that the centromeric distribution of CENP-A on mitotic human chromosomes is more complex than previously thought. CENP-A molecules are found on both the external and internal sides of centromeres. However, it remains unclear why p-CENP-AS7 molecules were never been detected on the inner side of the centromere. Peptide competition assays showed that the affinity of the CENP-A antibody (which is used in most CENP-A studies) for CENP-A was about five times higher than that for p-CENP-AS7 (Supplementary Fig. 1F). However, a fivefold difference in antibody affinities could be insufficient to account for the complete absence of p-CENP-AS7 detection by the anti-CENP-A antibody on the inner side of the centromere. Moreover,

the treatment of fixed chromosomes with Lambda phosphatase did not lead to the detection of CENP-A between sister kinetochores with anti-CENP-A antibodies (Supplementary Fig. 1G). These findings suggest that only a small amount of CENP-A is present on the inner side of the centromere, probably too little for detection by immunostaining with the CENP-A antibody. To investigate this hypothesis, we constructed two HeLa-S3 cell lines stably expressing hemagglutinin (HA)-tagged-CENP-A constructs resistant to a small interfering RNA (siRNA) targeting the 3′-untranslated region (UTR) of endogenous CENP-A, with and without mutation of the serine 7 residue (CENP-A-HA-S7A and CENP-A-HA-WT, respectively), to assess the relative proportion of the phosphorylated form of CENP-A (Supplementary Fig. 2). A single band corresponding to CENP-A-HA was observed in asynchronous cell extracts for both constructs (Fig. 1d). An additional low-intensity shifted band was specifically detected in mitotic CENP-A-HA-WT cell extracts and not in CENP-A-HA-S7A mitotic cell extracts (Fig. 1d). This shifted band results from the phosphorylation of serine 7 in mitotic CENP-A-HA-WT samples, given that the status of this residue is the only difference between the two cell lines. The signal for the shifted band is much weaker than that for total CENP-A, indicating that p-CENP-AS7 accounts for only a small proportion of the CENP-A in the extracts.

The CENP-A population recognized in immunofluorescence (IF) by the anti-CENP-A antibody therefore corresponds mostly to a non-phosphorylated form of CENP-A and is, thus, referred to here as "unphosphorylated CENP-A." Overall, our results reveal the presence of small amounts of inner centromere-bound p-CENP-AS7 molecules, occupying topological domains that, for the most part, do not overlap with those occupied by unphosphorylated CENP-A.

**p-CENP-AS7 is required for sister chromatid cohesion.** HeLa cells overexpressing a GFP-CENP-A-S7A construct display misaligned chromosomes and segregation defects[9]. Given its inner centromeric localization, p-CENP-AS7 may be a good candidate for chromatid cohesion-associated functions. We investigated

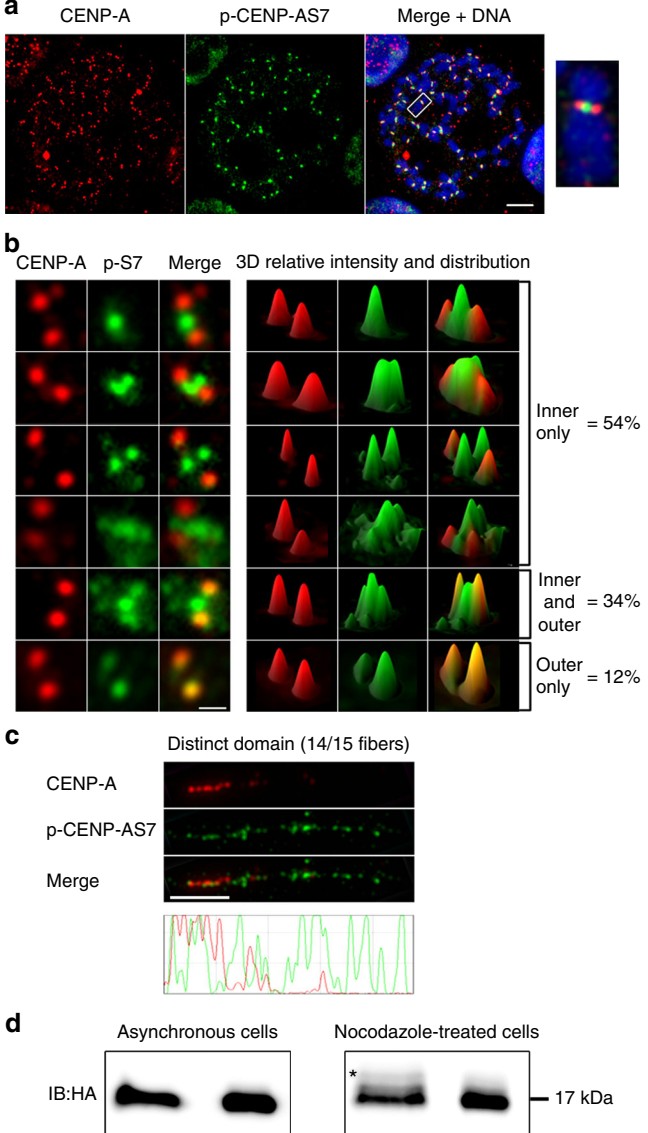

**Fig. 1** Serine 7-phosphorylated CENP-A (p-CENP-AS7) localizes to the inner side of centromeres during mitosis. **a** Immunofluorescence on metaphase chromosome spreads from HeLa-S3 cells, double-stained with anti-CENP-A and anti-p-CENP-AS7 antibodies. Merged image with DAPI staining (blue), scale bar = 10 μm. Magnification × 7.5 of a single chromosome. **b** Distribution of p-CENP-AS7 (green) and CENP-A (red) at centromeres. Left panels: different patterns of immunofluorescence signals with p-CENP-AS7 on the inner side only, on both the inner and outer sides and on the outer side only; right panels: 3D views of p-CENP-AS7 (green) and CENP-A (red) signals. Scale bar = 1 μm. In total, 520 chromosomes from 3 independent experiments were scored and the percentages of chromosomes with p-CENP-AS7 on the inner and/or outer side of the centromere are indicated on the right. **c** CENP-A and p-CENP-AS7 localize to different domains on mitotic stretched chromatin fibers. Extended chromatin fibers were prepared from HeLa-S3 cells blocked in mitosis by overnight nocodazole treatment; CENP-A (red) and p-CENP-AS7 (green) signals on a representative centromere chromatin fiber are shown. The lower panel shows the profile of signals along the length of the fiber. Scale bar = 5 μm. **d** Only a small amount of CENP-A displays serine 7 phosphorylation during mitosis. HeLa cell lines stably expressing CENP-A-WT-HA and CENP-A-S7A-HA were left untreated (left panel) or were treated overnight with nocodazole (right panel). Protein extracts were separated on a 20% polyacrylamide (29:1 acrylamide/bisacylamide) gel and immunoblotted with an anti-HA antibody. The shift due to phosphorylation is indicated by an asterisk. Three independent experiments were performed

whether the defects previously reported in GFP-CENP-A-S7A cells were the consequence of defective sister chromatid cohesion by studying chromosome spreads from CENP-A-HA-WT-, unphosphorylatable CENP-A-HA-S7A-, or phosphomimetic CENP-A-HA-S7D-expressing cells from which endogenous CENP-A was depleted. The cohesion phenotypes of cells were heterogeneous, with only a fraction of chromosomes in individual cells displaying PSCS. Only cells with clear phenotypes (i.e., separated chromatids for more than 50% of the chromosomes) were scored as PSCS. Sister chromatid cohesion was lost in 20% of the mitotic cells of the CENP-A-HA-S7A cell line, but maintained in CENP-A-HA-WT cells (Fig. 2). Cells expressing a phosphomimetic mutant (CENP-A-HA-S7D) do not display PSCS.

**Aurora A-dependent phosphorylation of CENP-AS7 protects cohesion.** Both Aurora A and Aurora B are responsible for CENP-AS7 phosphorylation during mitosis[9]. Aurora B is located at inner centromeres but its depletion does not lead to PSCS (Fig. 3a and Supplementary Fig. 3A for depletion controls). However, in CENP-A-HA-S7A cell lines, Aurora B is not recruited to centromeres (Supplementary Fig. 3B[9]), but remains active, as indicated by its capacity to phosphorylate H3S10 (Supplementary Fig. 3C[9]). We investigated whether a mislocalized, but active Aurora B was responsible for sister chromatid separation in CENP-A-HA-AS7A cells, using a cell line

expressing a Myc-tagged-Survivin D70A/D71A mutant[34]. The depletion of the endogenous wild-type (WT) Survivin in this mutant cell line has been shown to result in an inability of Aurora B to target centromeres (Supplementary Fig. 4A). However, Aurora B remains active, as indicated by immunostaining for p-H3S10 (Supplementary Fig. 4B) and p-CENP-AS7 (Supplementary Fig. 4C). In terms of Aurora B localization and activity, the conditions in this mutant are identical to those observed in CENP-A-HA-AS7A cells. In the Survivin-mutant cell line, the proportion of cells displaying PSCS was not significantly different from that in the control (Supplementary Fig. 4D). Thus, the absence of Aurora B localization to centromeres does not account for the PSCS observed in the CENP-A-HA-AS7A cell line.

Aurora A activity also contributes to CENP-AS7 phosphorylation during mitosis and its depletion leads to chromosome misalignments[9]. We investigated whether these defective alignments were due to a sister chromatid cohesion defect by depleting Aurora A from HeLa cells (Supplementary Fig. 3A). Depletion of Aurora A resulted in loss of p-CENP-AS7 (Fig. 3a). The lower level of CENP-A phosphorylation was linked to PSCS in 20% of mitotic cells (Fig. 3a). Similar results were also observed following the treatment of HeLa cells with a specific inhibitor of Aurora A (MLN8054; Fig. 3b). Inhibitor efficiency was monitored by assessing the loss of the active form of Aurora A (p-Aurora AT288). As in the Aurora A depletion experiments, about 20% of the mitotic cells displayed PSCS, demonstrating a role for the kinase activity of Aurora A in the protection of sister chromatid cohesion. The depletion of Aurora-A and the inhibition of its kinase activity lead to a loss of sister chromatid cohesion of similar magnitude (~20%) to that observed in CENP-A-HA-S7A cells. Furthermore, the inhibition of Aurora A in CENP-A-HA-S7A cells did not aggravate the PSCS phenotype observed in untreated CENP-A-HA-S7A cells, and expression of the phosphomimetic mutant CENP-A-HA-S7D abolished the chromatid separation induced by Aurora-A inhibition (Fig. 3c).

**Aurora A localizes to the centromere during mitosis.** Aurora A is known as a spindle-pole kinase involved in centrosome and spindle-related functions[35], but its role in the protection of sister chromatid cohesion at centromeres led us to investigate the possible recruitment of a previously unidentified fraction of Aurora A to the centromeres. We colabeled chromosome spreads with CREST serum and two different anti-Aurora A antibodies. Discrete Aurora A signals were clearly visible at centromeres after long exposure times (Fig. 4a). This signal was lost in cells transfected with an Aurora A siRNA, confirming the specificity of the Aurora A signal at centromeres (Fig. 4a). Specific signals were also detected at centromeres in a green fluorescent protein (GFP)-Aurora A-expressing cell line, following staining with an anti-GFP antibody (Fig. 4b). For confirmation of these results by biochemical approaches, CENP-A was co-immunoprecipitated with centromeric proteins, such as NDC80 and CENP-C, corresponding to the most distal and proximal CENP-A kinetochore interactors in vivo. Under these experimental conditions, a fraction of Aurora A was co-immunoprecipitated with CENP-A (Supplementary Fig. 4E, left panel). In a reverse experiment, we immunoprecipitated Aurora A from mitotic chromatin extracts and demonstrated the co-immunoprecipitation of a fraction of NDC80 and CENP-A (Supplementary Fig. 4E, right panel).

These results demonstrate the presence of endogenous Aurora A at centromeres and are consistent with previous results showing interactions between recombinant CENP-A and Aurora A[9].

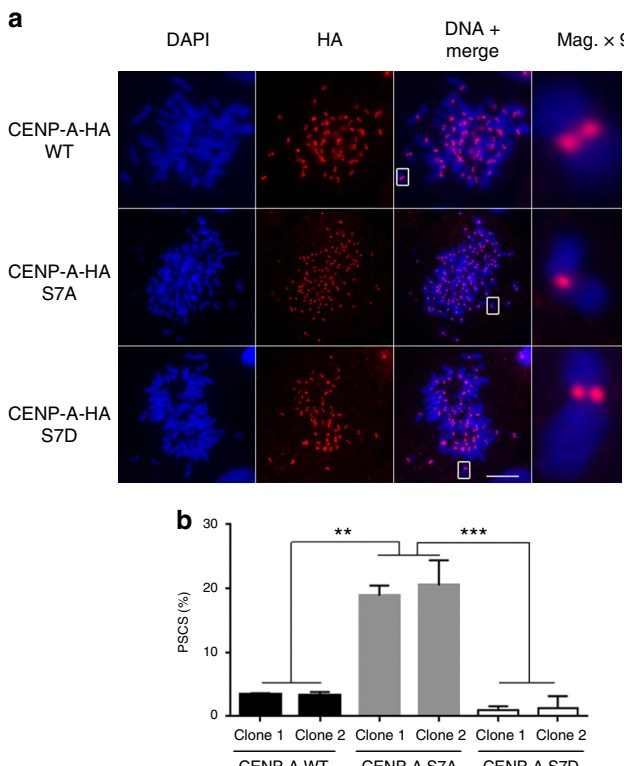

**Fig. 2** CENP-A-S7P is required for the protection of sister chromatid cohesion during mitosis. **a** After endogenous CENP-A siRNA depletion, metaphase chromosome spreads from CENP-A-HA-WT, CENP-A-HA-S7A, or CENP-A-HA-S7D cell lines were immunostained with an anti-HA antibody. A ×9 magnification of a single chromosome is shown. Scale bar = 5 μm. **b** Mitotic cells displaying premature sister chromatid separation (PSCS) were quantified with two different clones from each cell line; at least 1000 cells were counted for each data point. The data shown are the mean ± SD (n = 3 experiments; **P<0.01, ***P<0.001, one-way ANOVA)

**p-CENP-AS7 prevents cohesion loss in response to tension.** During mitosis, cohesins are removed from the chromosome arms by a phosphorylation-based mechanism known as the "prophase pathway," Sgo1 protecting cohesion at centromeres until the onset of anaphase. We therefore investigated the presence of Sgo1 at centromeres in CENP-A-HA-S7A cells depleted

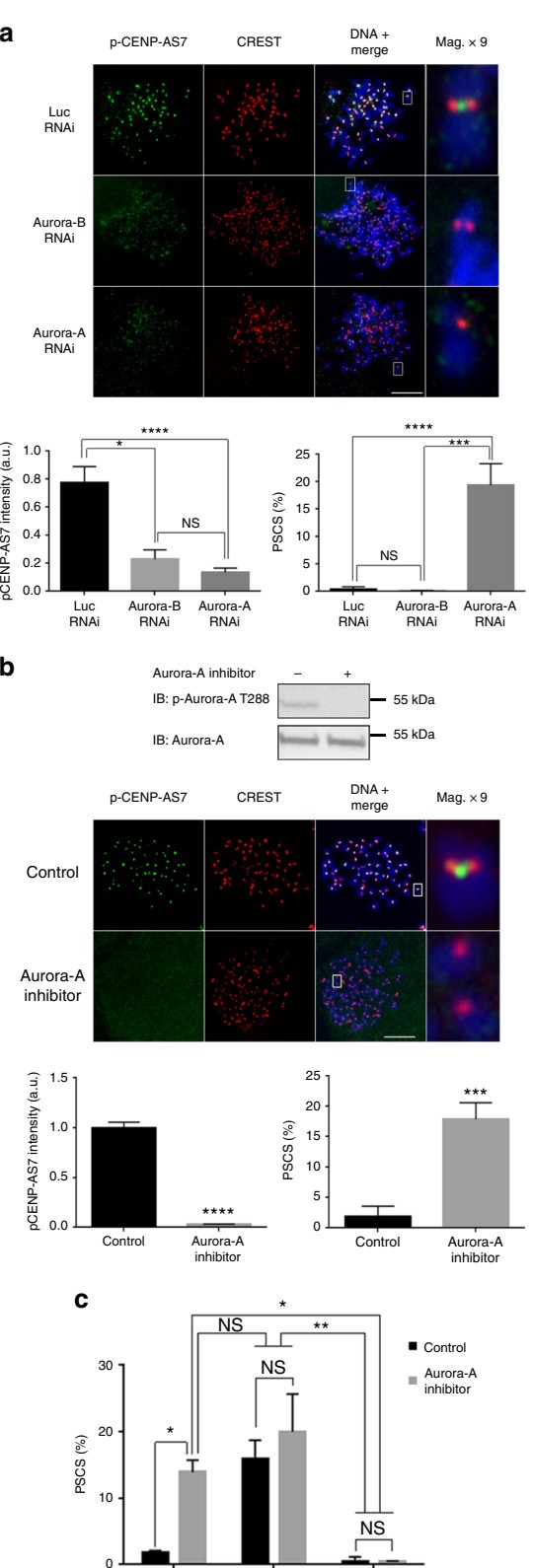

of endogenous CENP-A. Figure 5a (middle panels) shows two mitotic cells from the CENP-A-HA-S7A cell line. The sister chromatids are attached in the cell on the left (open arrow), whereas the cell on the right displays PSCS (closed arrow). CENP-A-HA-S7A cells displaying PSCS had lower levels of Sgo1 at centromeres than control WT cells, CENP-A-HA-S7D cells or CENP-A-HA-S7A cells displaying attached chromatids (Fig. 5a, b). Similar results were obtained in HeLa cells depleted of Aurora-A (Supplementary Fig. 5A). Conversely, Sgo1 depletion did not prevent CENP-AS7 phosphorylation or Aurora A recruitment to centromeres in cells displaying a loss of cohesion (Supplementary Fig. 5B and 5C). Moreover, the high levels of Sgo1 at centromeres in CENP-A-HA-S7A cells with joined sister chromatids showed that Sgo1 can still bind centromeres in the absence of CENP-AS7 phosphorylation.

Previous studies have shown that Sgo1 separates from cohesin in response to tension, thereby abolishing the protective function of Sgo1 against the prophase pathway[27]. We addressed the role of tension by treating CENP-A-HA-S7A cells with nocodazole for 3 h to disrupt the microtubule network, thereby preventing the establishment of tension. The number of cells displaying PSCS decreased markedly when tension was abolished and the Sgo1 signal remained strong at the centromeres of joined sister chromatids (Fig. 5b). Conversely, we treated cells with MG132 for 6 h to artificially stabilize biorientated chromosomes under tension. When tension was maintained in CENP-A-HA-S7A cells, the proportion of cells displaying PSCS was higher than that in the WT control (Fig. 5c). Furthermore, the Sgo1 signal on separated chromatids in MG132-treated CENP-A-HA-S7A cells was significantly weaker than that on separated chromatids from MG132-treated CENP-A-HA-WT control cells or from the MG132-treated CENP-A-HA-S7D cells (Fig. 5d). CENP-A depletion in HeLa cells mimics the expression of CENP-AS7A in terms of both the weakened resistance to tension and the lack

**Fig. 3** Aurora A is involved in sister chromatid cohesion. **a** After transfection with a siRNA against Aurora A or Aurora B, chromosome spreads were prepared from HeLa-S3 cells and immunostained with anti-p-CENP-AS7 and CREST antibodies. A × 9 magnification of a single chromosome is shown. The relative fluorescence intensity of p-CENP-AS7 relative to CREST was determined and mean values ± SEM are shown on the bottom left side of the panel ($n = 30$ cells for control and Aurora-A RNAi and 20 cells for Aurora-B RNAi, from a representative experiment; ****$P < 0.0001$; *$P < 0.1$; NS, not significant; non-parametric Kruskal–Wallis test). Bottom right panel: proportion of mitotic cells with PSCS ($n = 4$ experiments; > 200 cells per experiment were counted; ****$P < 0.0001$; ***$P < 0.001$; NS, not significant; one-way ANOVA) are shown. **b** HeLa cells were treated with MLN8054 (Aurora A inhibitor). Aurora A inhibition was monitored by western blotting with an anti-p-Aurora-AT288 as a marker of the Aurora-A active form (top of the panel); chromosome spreads were then stained with CREST (red) and anti-p-CENP-AS7 (green) antibodies. A × 9 magnification of a single chromosome is shown. Scale bar = 5 μm. The fluorescence intensity of p-CENP-AS7 signals relative to CREST signals mean values ± SEM are shown on the bottom left side of the panel ($n = 25$ cells for a representative experiment; ****$P < 0.0001$; non-parametric Kruskal–Wallis test). The mean ± SD of the percentage of cells displaying PSCS is shown on the bottom right of the panel ($n = 3$ experiments; > 200 cells per experiment were counted; ***$P < 0.001$; two-tailed unpaired parametric $t$ test). **c** Following endogenous CENP-A siRNA depletion, CENP-A-HA-WT, CENP-A-HA-S7A, or CENP-A-HA-S7D cell lines were treated for 1 h with 10 μM MLN8054 Aurora-A-specific inhibitor and then cytospun to determine the proportion of mitotic cell displaying PSCS. The data shown are the mean ± SD ($n = 3$ experiments; at least 200 cells were counted per condition; **$P < 0.01$, ***$P < 0.001$, two-way ANOVA)

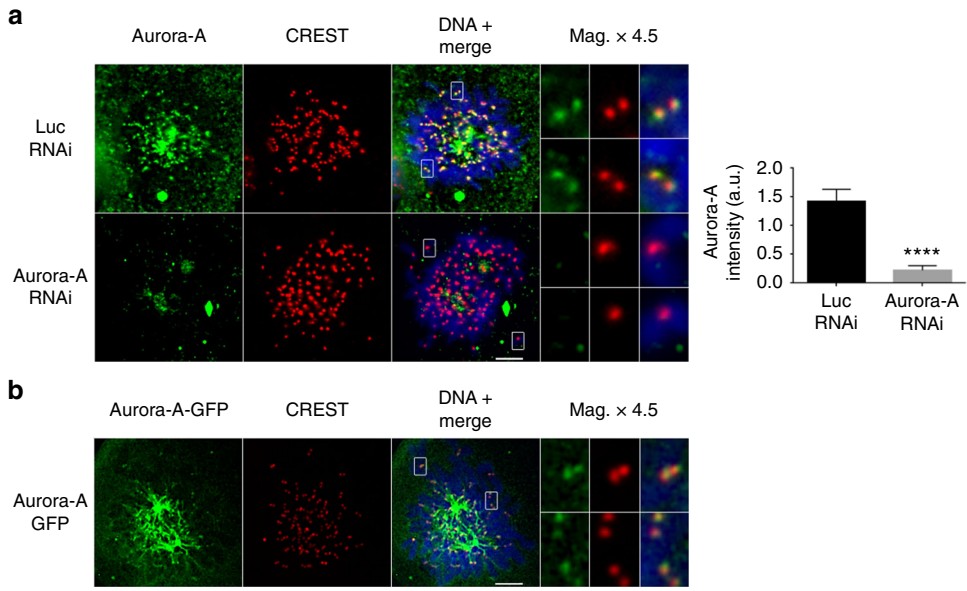

**Fig. 4** Aurora A localizes to the centromeres during mitosis. **a** Immunofluorescence microscopy on chromosome spreads, showing staining for Aurora A (green) and CREST (red) in HeLa cells transfected with control or Aurora-A siRNA. Images were overexposed to reveal weak signals. Magnifications (×4.5) of chromosomes, and mean ± SEM quantifications of centromeric fluorescence signals for Aurora A are shown ($n = 20$ cells from a representative experiment; ****$P < 0.0001$; two-tailed unpaired nonparametric Mann–Whitney $U$-test). **b** Immunofluorescence microscopy on chromosome spreads, showing staining for Aurora-A-GFP and CREST in U2OS cells stably expressing a GFP-tagged-Aurora-A. Magnifications of single chromosomes are shown. Scale bar = 5 µm

of Sgo1 stabilization on fatigue-induced separated chromatids, confirming the role of CENP-A in these two functions (Supplementary Fig. 6A-C). To ensure that the CENP-AS7A mutation does not cause an extended mitotic arrest in a metaphase-like stage that would eventually result in cohesion fatigue, cells were synchronized at the G2/M border, released for 30 min and supplemented with MG132. At the time of release, < 2% mitotic cells were observed (Fig. 5e, right graph) and we can, therefore, assume that, at a given time point after MG132 addition, both WT and S7A cells have spent identical amounts of time with their chromosomes under tension. However, following block release, the PSCS levels in CENP-A-HA-S7A cells were, reproducibly, twice those of control cells (Fig. 5e, left graph). These results are not consistent with the excess PSCS observed in CENP-AS7A cells being due to an extended mitotic block. Rather, they are consistent with a role for CENP-AS7 phosphorylation in resistance to cohesion fatigue. The accumulation of untreated CENP-A-HA-S7A cells in mitosis after block release (Fig. 5e, right graph) is, therefore, probably due to SAC activation in response to cohesion defects. Strikingly, cells expressing a phosphomimetic CENP-AS7D construct were more resistant to sustained tension than the WT control (Fig. 5c) and, on the few separated sister chromatids observed, Sgo1 intensity had not decreased as markedly as in CENP-AS7A cells displaying PSCS (Fig. 5d). We therefore conclude that p-CENP-AS7 helps to protect against premature cohesion loss at centromeres when spindle force-induced tension is established during chromosome alignment.

**Bub1 is required for Aurora A recruitment at centromeres**. In human cells, after the bipolar attachment of sister chromatids, Sgo1 is relocated from the inner to the outer centromere through binding to H2AT120 that has been phosphorylated by Bub1[27, 28]. In budding yeast, Bub1 elimination leads to the removal of Sgo1 homologs in response to tension, but this effect is independent of

H2AS121 (corresponding to H2AT120 in humans) phosphorylation[30]. We addressed the question of a possible link between Bub1 and the role of p-CENP-AS7 in Sgo1 localization when chromosomes are under tension. In CENP-A-HA-S7A cells, Bub1 is still targeted to centromeres in cells displaying PSCS (Supplementary Fig. 7A) and H2AT120 phosphorylation is not affected (Supplementary Fig. 7B). Accordingly, Aurora A depletion did not prevent Bub1 recruitment to the separated sister chromatids (Supplementary Fig. 7A) or H2AT120 phosphorylation (Supplementary Fig. 7B). By contrast, Bub1 depletion prevented CENP-AS7 phosphorylation (Fig. 6a) and Aurora A recruitment to centromeres (Fig. 6b). Bub1 depletion leads to the loss of H2AT120 phosphorylation, the delocalization of Sgo1 from centromeres and a 1.3 times increase in inter-kinetochore distance. However, due to residual chromosome arm cohesion, Bub1-depleted cells do not display PSCS[36]. We investigated the phenotype of cells expressing mutant forms of CENP-AS7 lacking Bub1 (Fig. 6c). As previously shown by Kitajima et al.[36], inter-kinetochore distance increased by a factor of 1.25 upon Bub1 depletion in CENP-A-HA-WT cells (1.15 µm vs. 0.9 µm; Fig. 6c). CENP-A-HA-S7A cells display no such plasticity, instead retaining their "open" conformation (inter-kinetochore distance of about 1.15 µm) regardless of Bub1 status. Conversely, cells expressing the CENP-A-HA-S7D mutant adopt a "closed" conformation (inter-kinetochore distance of about 0.9 µm) in both the presence and absence of Bub1. Thus, inter-kinetochore distance seems to be under the control of CENP-AS7 phosphorylation. No Sgo1 signal was detected at centromeres in Bub1-depleted CENP-A-HA-S7D cells (Fig. 6c, lower panels), confirming that p-CENP-AS7 is not a primary docking site for Sgo1, despite its role in retaining Sgo1 on separated sister chromatids.

## Discussion

Current models describing chromatin organization at centromeres in mitotic chromosomes restrict CENP-A

nucleosomes to the outer centromere, with canonical H3 nucleosomes located on the inner side. Our results show that CENP-A molecules occupy larger chromosome domains than expected, on both the external and internal sides of

centromeres. We found that only a small proportion of total CENP-A displayed serine 7 phosphorylation. This may contribute to explain the lack of detection of p-CENP-AS7 by mass spectrometry[10].

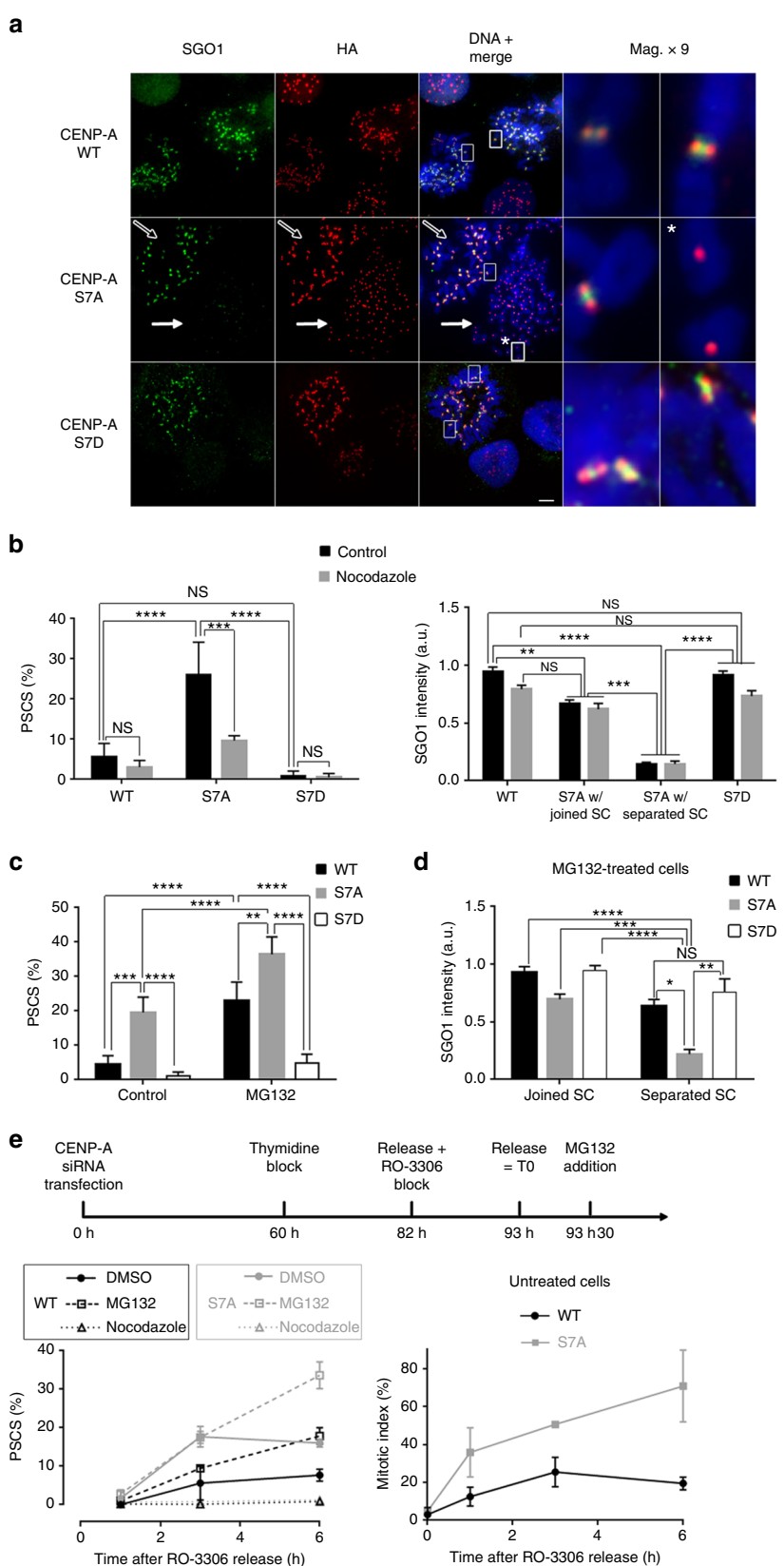

In addition to being localized at inner centromeres, p-CENP-AS7 was not restricted to the constriction point during mitosis, instead often spilling over into adjacent regions and displaying various distributions patterns, whereas unphosphorylated CENP-A signals were always clearly restricted to the outer centromere. Despite accounting for the vast majority of total CENP-A chromatin, the unphosphorylated CENP-A chromatin domains seemed to occupy a shorter length of stretched mitotic CFs than p-CENP-AS7 domains. These results indicate that p-CENP-AS7-containing chromatin may be more stretchable and, possibly, more dynamic than its unphosphorylated counterpart.

The centromere propagates epigenetically through cell generations via the incorporation of CENP-A nucleosomes. However, as CENP-A was thought to be restricted to the outer centromere, it remains unclear how the information about "being a centromere" is carried to the inner centromere. The detection of CENP-A at inner centromeres in this study suggests that CENP-A may act as an inner centromere epigenetic mark for inheritable inner centromere-specific functions, and they also imply that current models of centromere chromatin organization are in need of reconsideration.

The finding of a role for Aurora A in centromeric functions was unexpected, because this kinase has been reported to be associated with spindle poles and involved in centrosome maturation and spindle-related functions. However, several lines of evidence suggest that Aurora A can phosphorylate centromere substrates. For example, Aurora A has been reported to be involved in microtubule nucleation in the vicinity of the kinetochore/chromatin and in the formation and stabilization of microtubules[37]. Kinetochore proteins, such as NDC80 or CENP-E, have been shown to be substrates of Aurora A, and Aurora A-mediated phosphorylation of histone H3T118 at pericentromeres is required for correct chromosome segregation[38–40]. Aurora A is also involved in SAC activation in response to microtubule-perturbing agents[41]. In mouse oocytes, Aurora A participates in the correction of defective kinetochore-microtubule attachment. An overexpressed EGFP-Aurora-A can be detected at kinetochores and this overexpression leads to the destabilization of kinetochore-microtubule attachments during meiosis I[42]. However, the endogenous Aurora A was not detected at centromeres and the authors suggested that, in the absence of EGFP-Aurora-A overexpression, the Aurora A-mediated correction of kinetochore–microtubule attachments is restricted to chromosomes located close to spindle poles[42]. Our demonstration that a fraction of Aurora A localizes to the kinetochores of mitotic cells

may explain the existence of a number of Aurora A substrates at centromeres, without requiring chromosomes to be located close to centrosomes or spindle poles.

Both Aurora A and Aurora B phosphorylate CENP-A-S7 during mitosis, but only Aurora A depletion leads to sister chromatid cohesion defects. The function of CENP-AS7 phosphorylation by Aurora B therefore remains unclear. However, the depletion of Aurora B or the inhibition of its kinase activity leads to a loss of amphitelic attachment, preventing chromosome biorientation and tension establishment, and we show here that tension is required for CENP-AS7A-dependent PSCS. The possible role of Aurora B in protecting centromere cohesion against sustained tension at centromeres would thus be masked by its role in kinetochore/microtubule attachment processing. Moreover, Aurora B has been identified as involved in the "prophase pathway" of cohesin removal, and its inhibition or depletion would reinforce sister chromatid cohesion, making it impossible to assess its potential role in resistance to fatigue[43]. By contrast, this study demonstrates the role of Aurora A in CENP-AS7 phosphorylation and the subsequent resistance to cohesion fatigue. Indeed, the depletion or inhibition of Aurora A phenocopies CENP-AS7A expression, and CENP-AS7D cells do not display PSCS, even upon Aurora A inhibition.

Cohesion fatigue is a phenomenon that leads to uncoordinated sister chromatid disjunction when chromosomes are subjected to sustained tension[31, 32]. We report here that the expression of an unphosphorylatable mutant of CENP-AS7 leads to cohesion fatigue. It has been reported that overexpression of CENP-AS7A leads to chromosome alignment defects and the authors suggested that this defect would result from improper microtubule attachment to kinetochores[9]. However, chromosome misalignment is a known phenotype associated with unscheduled sister chromatid separation[31, 32]. Therefore, the simplest interpretation is that, in CENP-AS7A-expressing cells, alignment defects would be the consequence of cohesion fatigue rather than of a failure to connect microtubules to kinetochores. It would be tempting to interpret the cohesion defects as a consequence of mutant-induced extended mitotic arrest, with chromosomes under tension. However, assays involving cell synchronization at the G2/M border showed that cells expressing a mutant CENP-A-HA-S7A construct were less resistant to sustained tension than CENP-A-HA-WT cells, although neither of these cell types was yet in mitosis after block release and both cell types had spent the same amount of time under tension following entry into mitosis (Fig. 5e). Moreover, we found that a phosphomimetic mutant

**Fig. 5** CENP-AS7 phosphorylation is involved in sister chromatid cohesion and Sgo1 maintenance at centromeres in response to tension. **a** Chromosome spreads from CENP-A-HA-WT, CENP-A-HA-S7A, and CENP-A-HA-S7D cells depleted of endogenous CENP-A were immunostained for Sgo1 and HA. Magnifications (× 9) are shown. In the CENP-A-HA-S7A cell line, the presence and absence of Sgo1 immunostaining at the centromere of joined (empty arrow) or separated sister chromatids (full arrow and asterisk), respectively, are shown. Scale bar = 5 μm. **b** CENP-A-HA-WT and CENP-A-HA-S7A cell lines were treated or not with nocodazole for 3 hours. The means ± SD of PSCS percentages are shown on the left (n = 2 experiments; > 150 cells per experiment were counted; ****$P < 0.0001$; ***$P < 0.001$; NS, not significant; two-way ANOVA). Quantifications of Sgo1 in cell lines presenting joined or separated sister chromatids (SC) are shown on the right. The graph shows the means ± SEM from a representative experiment (n = at least 22 cells; ****$P < 0.0001$; ***$P < 0.001$; **$P < 0.01$; NS, not significant; non-parametric Kruskal–Wallis test). **c** CENP-A-HA-WT, CENP-A-HA-S7A and CENP-A-HA-S7D cell lines depleted from endogenous CENP-A were treated with 20 μM MG132 for 6 h and PSCS were scored. Means ± SD for the proportion of cells displaying PSCS are shown (n = 4 experiments; > 150 cells per experiment were counted; ****$P < 0.0001$; ***$P < 0.001$; **$P < 0.01$; two-way ANOVA). **d** Means ± SEM from a representative experiment quantifying Sgo1 fluorescence intensity on joined or separated SC after the depletion of endogenous CENP-A in CENP-A-HA-WT, CENP-A-HA-S7A, and CENP-A-HA-S7D cell lines further treated with 20 μM MG132 for 6 h (n = 23–26 cells, except for S7D with separated SC where n = 14; ****$P < 0.0001$; ***$P < 0.001$; **$P < 0.01$; *$P < 0.1$; NS, not significant; non-parametric Kruskal–Wallis test). **e** CENP-A-HA-WT and CENP-A-HA-S7A cell lines depleted from endogenous CENP-A and synchronized as indicated were scored for PSCS after 3 or 6 h following nocodazole or MG132 treatment. The means ± SD of PSCS percentages are shown on the left (n = 2 experiments; > 200 cells per experiment). Right panel: mitotic indexes (n = 2 experiments, > 400 cells per experiment)

(CENP-A-HA-S7D) was significantly more resistant to cohesion fatigue than a WT control (Fig. 5c). These data show that the mitotic arrest observed in CENP-A-HA-S7A-expressing cells is a consequence rather than a cause of greater cohesion fatigue. The accumulation of untreated S7A cells in mitosis after block release (Fig. 5e right graph) is, therefore, probably due to SAC activation in response to cohesion defects. Strikingly, cells expressing a phosphomimetic CENP-A-HA-S7D construct were more

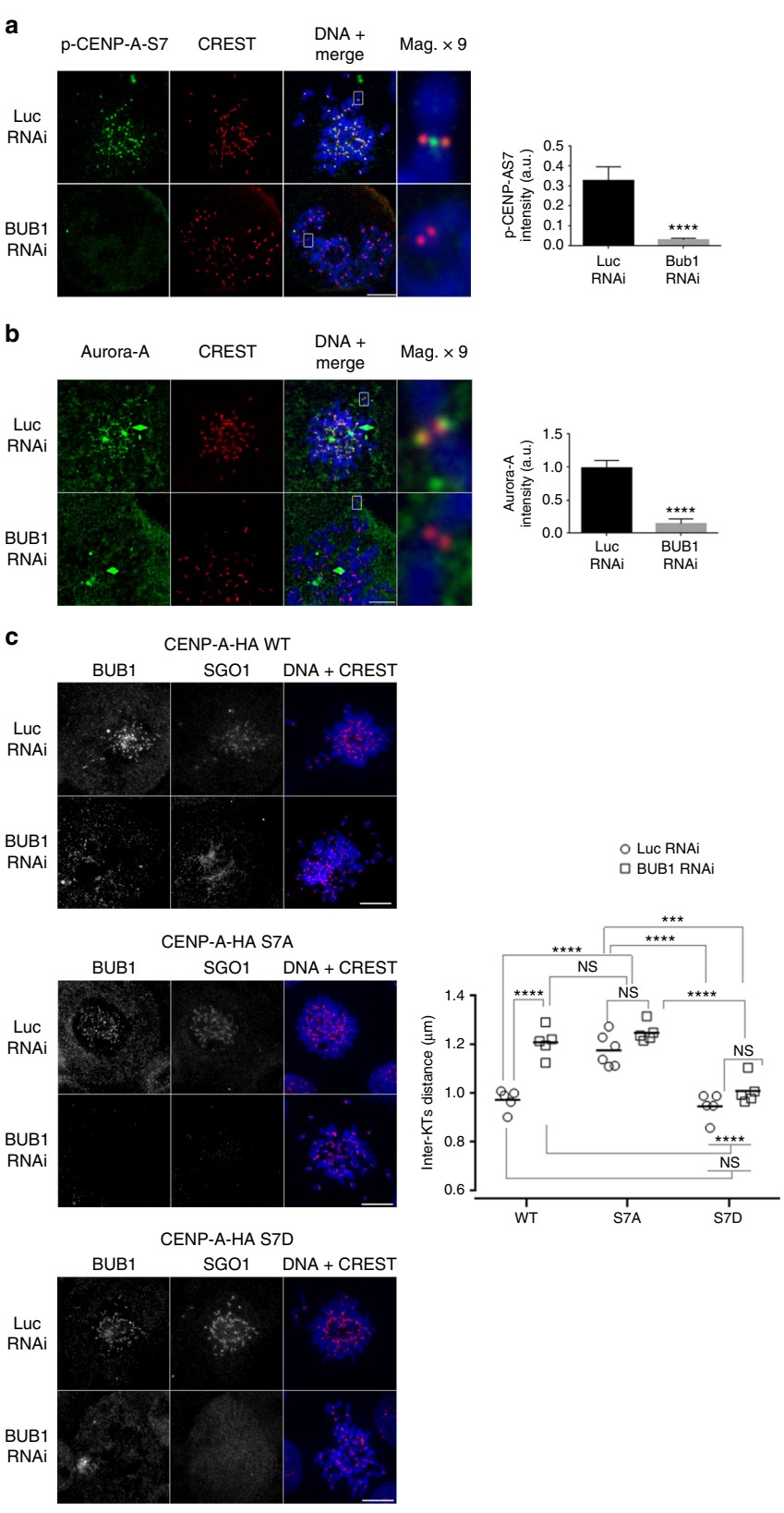

resistant to sustained tension than the WT control (Fig. 5c) and, on the few separated sister chromatids observed, Sgo1 intensity had no decreased to as great an extent as in CENP-A-HA-S7A cells displaying PSCS (Fig. 5d). We therefore conclude that p-CENP-AS7 helps to protect against premature cohesion loss at centromeres when spindle force-induced tension is established during chromosome alignment.

The combination of Aurora A inhibition with CENP-A-HA-S7A expression did not lead to more PSCS than Aurora A inhibition or CENP-A-HA-S7A expression alone. Thus, Aurora A and p-CENP-AS7 appear to be belong to the same epistasis group for PSCS phenotype. Moreover, expression of the phosphomimetic mutant CENP-A-HA-S7D rescues the cohesion defects induced by Aurora A inhibition. This, together with the observation that CENP-AS7 is an Aurora A substrate in vitro[9], indicates that Aurora A-mediated CENP-AS7 phosphorylation protects cohesion against fatigue.

We report that the prevention of CENP-AS7 phosphorylation leads to cohesion fatigue and the partial removal of Sgo1 from the centromeres of separated chromatids. It was not possible to demonstrate a causal relationship between these two observations, but cohesion defects are unlikely to be the sole cause of Sgo1 loss from centromeres since cohesin depletion do not lead to Sgo1 removal on separated chromatids (Supplementary Fig. 8)[28, 44]. Thus, Sgo1 loss from the centromeres of separated chromatids in CENP-A-HA-S7A cells seems to be the consequence of both tension (because MG132-treated CENP-A-HA-WT cells have significantly less Sgo1 on separated chromatids than on attached chromatids) and the S7A mutation, because the decrease in the Sgo1 signal on both attached and unattached chromatids is more pronounced in CENP-A-HA-S7A cells than in CENP-A-HA-WT cells (Fig. 5b,d). In Saccharomyces cerevisiae, when chromosomes are correctly biorientated, Bub1 removal from centromeres leads to Sgo1 eviction, independently of H2AS121 (corresponding to H2AT120 in human cells) phosphorylation[30]. This is consistent with our observation that the localization of Aurora A to centromeres and CENP-AS7-associated phosphorylation are controlled by Bub1. Our data suggest that, in addition to phosphorylating H2AT120, Bub1 reinforces the protection of sister chromatid cohesion by maintaining Sgo1 at centromeres through the recruitment of Aurora A and CENP-A-S7 phosphorylation.

A role of p-CENP-A-S7 as a docking site for Sgo1 would be an attractive hypothesis. However, we showed that CENP-AS7A mutation did not prevent Sgo1 recruitment, but weakened the anchoring of this protein to centromeres only when the sister chromatids were under tension. Furthermore, this hypothesis is not supported by the observation that expression of the phosphomimetic mutant CENP-AS7D in Bub1-depleted cells does not rescue Sgo1 localization (Fig. 6c). Finally, given that the sequence of the N-terminal part of CENP-A is not conserved, it seems unlikely that it acts as a docking site for a conserved protein. Alternatively, the phosphorylation of CENP-AS7 may modify

inner centromere chromatin plasticity, as suggested by the results obtained for chromosome spreads and stretched mitotic CFs. This hypothesis is supported by the observation that inter-kinetochore distance is under the control of CENP-AS7 status (Fig. 6c).

In human cells, the distribution of Sgo1 at centromeres depends on whether tension has been established between sister kinetochores[27, 28]. When chromosomes are not bioriented, Sgo1 localizes at inner centromeres where it interacts with cohesin[23]. By contrast, when tension is established, Sgo1 is redistributed to the outer centromeres, where it colocalizes with p-H2AT120. Interestingly, a mutant of Sgo1 that binds cohesin but has lost the ability to bind p-H2AT120 does not rescue cohesion fatigue, whereas WT Sgo1 does[28]. These data indicate that, following its tension-induced relocation to p-H2AT120 at the outer centromere, Sgo1 is involved in protection against cohesion fatigue. The loss of Aurora A and the mutation of CENP-AS7 do not modify H2AT120 phosphorylation (Supplementary Fig. 7B). Moreover, following sustained tension, separated sister chromatids in CENP-A-HA-S7A cells have a lower Sgo1 signal intensity than separated sister chromatids in CENP-A-HA-WT cells (Fig. 5d). We therefore favor a model in which p-CENP-AS7 indirectly stabilizes Sgo1 at centromeres after the establishment of tension to protect against fatigue.

The amphitelic attachment of microtubules to kinetochores induces chromosome biorientation. Sister centromeres are thus placed under tension and Sgo1 relocates to the outer centromere. However, the SAC ensures that the cell cannot proceed to anaphase until all the chromosomes are bioriented, and cohesion must therefore continue to be protected. We propose that, during this short time window, when some chromosomes are bioriented but others are not yet under tension, Aurora A-dependent CENP-AS7 phosphorylation consolidates Sgo1 at centromeres, providing protection against fatigue until the last chromosome is bioriented and the SAC is satisfied.

## Methods

**Antibodies.** Antibodies dilutions for western blotting (WB), IF, and CFs are indicated below. The antibodies used were as follows: mouse monoclonal anti-CENP-A (Abcam 13939) WB: 1/1000, IF: 1/100; rabbit monoclonal anti-p-CENP-AS7 (Millipore 04-792) WB: 1/1000, IF: 1/1000, CF: 1/100; rabbit polyclonal anti-p-CENP-AS7 (Millipore 07-232) IF: 1/100; rabbit polyclonal anti-p-CENP-AS7 (Cell Signaling Technologies 2187) IF: 1/100; rat monoclonal anti-HA (Roche 2013819) (WB: HA-HRP 1/500 Roche 20138, IF: anti-HA Rat Roche 1867423 1/100); mouse monoclonal anti-Aurora B (BD Transduction 611083) IF: 1/200, WB: 1/1000; mouse monoclonal anti-p-H3S10 (Millipore 05-806) IF: 1/50,000, CF: 1/1000; rabbit polyclonal anti-p-H3S10 (Millipore 06-570) IF: 1/1000, CF: 1/200; rabbit polyclonal anti-Aurora A (Abcam Ab12875) IF: 1/100; mouse monoclonal anti-Aurora A (Abcam 13824) IF: 1/100, WB: 1/1000; rabbit polyclonal anti-actin (Sigma A5060) WB: 1/10,000; rabbit polyclonal anti-Survivin (Abcam 469) WB: 1/5000; mouse monoclonal anti-Bub1 (Sigma B0561) IF: 1/200; and rabbit polyclonal anti-p-H2A-T120 (Active Motif 39392) IF: 1/1000. Affinity-purified rabbit anti-Sgo1 antibody was a gift from Professor Yoshinori Watanabe (Japan) WB and IF: 1/1000; Human CREST serum was a gift from Dr Isabelle Bahon-Riedinger (France) IF: 1/4000.

**Fig. 6** Bub1 regulates CENP-AS7 phosphorylation and Aurora-A targeting at centromeres. **a, b** HeLa cells depleted of Bub1 by siRNA were immunostained with anti-p-CENP-AS7 (**a**) or anti-Aurora-A (**b**) and CREST antibodies. Magnifications (×9) of single chromosomes are shown. Means ± SEM are shown (on the right of the panel) for the quantification of signal intensity relative to CREST obtained with antibodies against p-CENP-AS7 or Aurora A antibodies in a representative experiment (for p-CENP-AS7 intensity, n = 20 cells in each condition; for Aurora A intensity, n = 25 cells in control and 20 cells in Bub1 RNAi conditions; ****P < 0.0001; two-tailed unpaired nonparametric Mann–Whitney U-test). **c** Metaphase chromosome spreads from CENP-A-HA-WT, CENP-A-HA-S7A, and CENP-A-HA-S7D cell lines depleted of Bub1 were immunostained with antibodies against Bub1 and Sgo1. Scale bar = 5 µm. Inter-kinetochore distance means ± SEM of five to six representative cells with joined SC are shown (n = 10–50 chromosomes for each cell). Statistical analysis were performed on the mean of the distance measured for each cell (n = 5–6; ****P < 0.0001; ***P < 0.001; NS, not significant; two-way ANOVA with Tukey's post hoc analysis)

Alexa Fluor-coupled secondary antibodies from Invitrogen (IF: 1/1000) and horseradish peroxidase (HRP)-coupled secondary antibodies (Jackson Immunoresearch) (WB: Mouse 1/5000, Rabbit 1/25,000) were used for detection.

**Cell culture**. The cells used in this study were as follows: HeLa-S3 cells (ATCC CCL-2.2); U2OS cells stably expressing a GFP-tagged-Aurora A protein (A gift from Dr M. -B. Troadec); and HeLa cells stably expressing a Myc-tagged-Survivin-WT or a Myc-tagged-Survivin-D70A/D71A, in which the D70 and D71 residues of Survivin were mutated to alanine residues, preventing CPC binding to p-H3T3 and thus its recruitment to centromeres (a gift from Professor J. Higgins)[34]. The CENP-A-HA-WT, CENP-A-HA-S7A, and CENP-A-HA-S7D cell lines were obtained by stably transfecting HeLa-S3 cells with an expression vector containing the complementary cDNA of CENP-A tagged in 3′ with two successive HA sequences (CENP-A-HA-WT), the same construct with CENP-A serine 7 mutated to an alanine residue, preventing phosphorylation at this position (CENP-A-HA-S7A), or the same construct with CENP-A serine 7 mutated to an aspartate residue, mimicking phosphorylation at this position (CENP-A-HA-S7D). In these constructs, the 3′-UTR sequence corresponding to the siRNA target was scrambled to make these constructs resistant to downregulation of the endogenous CENP-A by RNA interference. Transfected stable cell lines were cloned by limiting dilution and individual clones expressing HA-tagged CENP-A in amounts similar to those for endogenous CENP-A were selected for further studies. In most experiments using the CENP-A-HA WT or S7A cell lines, expression of the endogenous *CENP-A* gene was silenced by two transfections, at 48 h intervals, with a siRNA directed against the 3′-UTR region of the CENP-A messenger RNA, with incubation for an additional 48 h before analysis. All cell lines were cultured in Dulbecco's modified Eagle's medium supplemented with 10% fetal bovine serum, 0.03% L-glutamine, 100 U/ml penicillin, and 100 μg/ml streptomycin. We added 1 mg/ml G418 to the culture medium of stably transfected cell lines. For synchronization experiment (Fig. 5e), after a 60 h incubation following CENP-A siRNA transfection, cells were blocked for 22 h with 2 mM thymidine, washed two times with 1 × phosphate-buffered saline (PBS) for 3 min and released in complete medium supplemented with 10 μM RO-3306 (CDK1 inhibitor). After 11 h of incubation, cells were washed with 1 × PBS as previously described and released in complete medium for 30 min. Then, MG132 and nocodazole were added at a final concentration of 10 μM and 100 ng/ml, respectively.

**siRNA transfection**. Cell lines were transfected with siRNA using Hiperfect transfection reagent (Qiagen), according to the manufacturer's instructions. Cells were transfected once or twice, within a 24- or 48 h period and then analyzed 1, 2, or 3 days after the second transfection, depending on the targeted gene. The following siRNAs (Qiagen or Dharmacon) were used: CENP-A siRNA: 5′-TAGCTCTTTCTGTAATATTTA-3′; Aurora-B siRNA: 5′-CGCGGCACUUCA-CAAUUGAdTdT-3′[45]; Aurora-A siRNA-1: 5′-AUGCCCUGUCUUACUGU-CAdTdT-3′[46]; Sgo1 siRNA: 5′-GUCUACUGAUAAUGUCUUAUT-3′[36]; Survivin siRNA: 5′-GCAGGUUCCUUAUCUGUCAdTdT-3′[47]; Bub1 siRNA: 5′-GGUUAUUUCAGACACGCCUUT-3′ (Ambion validated siRNA); and SCC1 siRNA: 5′- CGAUGAGCCCAUUAUUGAAdTdT-3′ (Qiagen validated siRNA).

**Cell treatments**. Peptide competition was achieved with various amounts (as indicated in the Supplemental Figures) of either CENP-A (GPRRRSRKPEAPRRR) or p-CENP-AS7 (GPRRR-pS-RKPEAPRRR) peptides, corresponding to amino acids 2-16 of CENP-A protein. These peptides were incubated with anti-CENP-A or anti-CENP-AS7 antibodies for 1 h at room temperature in 50 μl 5% fetal calf serum (FCS) in PBS before IF staining. Control peptide competition was carried out with 1 μg H4K5ac (SGRG-KAc-GGKGLGKGGA) peptide.

Fixed chromosome spreads obtained by cytocentrifugation were incubated with 5 units of Lambda phosphatase (BioLabs P0753S) for 2 h at 30 °C, washed three times in PBS, blocked with 5% FCS, and immunostained with the antibodies indicated (Supplemental Figures).

For mitotic CF preparations, HeLa-S3 cells were treated for 16 h with nocodazole (50 ng/ml) and mitotic cells were collected by shake-off. Cells were then subjected to hypotonic treatment with 75 mM KCl for 15 min at room temperature and 2000 cells were cytocentrifuged on slides for 5 min at 900 r.p.m. in a cytospin-4 (Thermo Scientific). Slides were incubated vertically for 30 min at room temperature in fiber buffer: 25 mM Tris pH 8, 1 M urea, 1% Triton, 30% glycerol, 500 mM NaCl, 10 mM EDTA. The slides were then slowly removed from the buffer and CFs were fixed and immunostained as described below. Despite our efforts to optimize this method, only one exploitable stretched mitotic centromere CF was obtained for every 10 slides carefully examined, on average. For inhibition of Aurora A kinase activity, HeLa-S3 cells were treated with 10 μM Aurora A inhibitor (MLN8054) for 1 h before collecting.

**Chromatin immunoprecipitation**. HeLa cells were prepared by pelleting and resuspending cells in 2.5 PCVs (pellet cell volumes) of wash buffer containing 20 mM HEPES pH 8, 1.5 mM MgCl$_2$, 20 mM KCl, 0.34 M sucrose, 10% glycerol, 0.5 mM dithiothreitol, phosphatase inhibitors (10 mM sodium butyrate, 10 mM β-glycerophosphate, 1 mM sodium pyrophosphate, 5 mM NaF, 0.2 mM Na$_3$Vo$_4$) and

protease inhibitor cocktail (Complete EDTA-free Ultra tablets from Sigma-Aldrich, used according to the manufacturer's instructions). The mixture was completed with 2.5 PCVs of wash buffer supplemented with 0.2% IGEPAL CA-630 and incubated on ice for 10 min. Nuclei were pelleted by centrifugation at 300 × g for 5 min, and the pellet was washed once in 5 PCVs of the same buffer and then once in 1 PCV of the same buffer supplemented with 300 mM NaCl. Nuclei were resuspended in 1 PCV of wash buffer supplemented with 300 mM NaCl and 3 mM CaCl. Chromatin was digested for 20 min at 37 °C with 5 units/ml micrococcal nuclease (Sigma ref: N3755). The extracts were supplemented with 5 mM EGTA, 0.5 mM EDTA, and 0.1% IGEPAL CA-630, incubated for 10 min on ice and centrifuged at 13,000 × g for 10 min at 4 °C. This first supernatant was conserved and the pellet was incubated with 0.5 PCVs of 1 mM EDTA, 0.1% IGEPAL, phosphatatase inhibitors, and protease inhibitor cocktail for 45 min at 4 °C, with shaking. We then added 0.5 PCVs of 40 mM HEPES pH 7.5, 600 mM NaCl, 0.1% IGEPAL, 20% glycerol, phosphatase inhibitors and protease inhibitor cocktail and incubated the mixture for another 15 min, with shaking. The cellular debris was removed by centrifugation for 10 min at 13,000 × g and 4 °C. This second supernatant was mixed with the first and used as the starting material for overnight immunoprecipitation with an in-house anti-CENP-A antibody raised in goat or rabbit anti-Aurora-A (Abcam ; ref: Ab12875). Purified centromeric chromatin was isolated by incubating the mixture with magnetic protein G Dynabeads (ThermoFisher Scientific), which were then washed three times with 20 mM HEPES pH 7.5, 300 mM NaCl, 0.5 mM EDTA, 0.1% IGEPAL, 10% glycerol, phosphatase inhibitor, before elution with the peptide used for immunization (0.8 mg/ml) for 1 h at 4 °C. We added 5 × SDS-polyacrylamide gel electrophoresis (PAGE) loading buffer to the eluted fractions, which were then subjected to WB analysis.

**Western blotting**. For immunoblotting, cells were lysed in Laemmli buffer (10% glycerol, 1.2% SDS, 0.02% bromophenol blue, 2% β-mercaptoethanol) and boiled for 5 min. For mitotic cell extracts, cell lines were treated overnight with 200 ng/ml nocodazole (Sigma M1404) before lysis. Samples were then subjected to SDS-PAGE in a 4–20% polyacrylamide gradient gel (Biorad) and transferred to polyvinylidene difluoride (PVDF) membranes (Biorad). Membranes were incubated overnight at 4 °C with primary antibodies and then for 1 h at room temperature with HRP-conjugated secondary antibodies, according to standard procedures.

Phosphorylated forms of CENP-A from CENP-A-HA-WT and CENP-HA-AS7A cell lines were separated on a 16 cm 20% polyacrylamide (29:1 acrylamide/bisacrylamide) gel. Electrophoresis was carried out in Tris-glycine buffer for 16 h at 4 °C. The resulting bands were transferred to PVDF membranes (Bio-Rad) and immunoblotted with anti-HA antibodies. WB signal detection was performed on a Luminescent Image Analyzer–Image Quant Las 4000 mini (GE HealthCare), according to the manufacturer's instructions. Results were obtained three times in triplicate. Uncropped gel scans are shown in the Supplementary Information (Supplementary Fig. 9 and 10).

**Immunofluorescence**. Chromosome spreads were obtained by cytocentrifugation. Briefly, cells (with or without treatment) were collected and subjected to hypotonic shock with 75 mM KCl for 15 min at room temperature. We then cytospun 30,000–50,000 cells on slides for 5 min at 900 r.p.m. (Cytospin 4; Thermo Scientific) before fixing them in 3% paraformaldehyde. Slides were then washed three times in PBS, permeabilized with 0.1% Triton X-100, washed three times in PBS, and blocked by incubation with 5% FCS. Slides were then incubated overnight at 4 °C with primary antibodies, washed five times with PBS, and then incubated for 1 h at room temperature with fluorochrome-conjugated secondary antibodies. The DNA was stained with DAPI (4',6-diamidino-2-phenylindole) (100 ng/ml in PBS) and slides were mounted in Prolong-Gold medium (Invitrogen).

**Microscopy and image analysis**. Images were acquired with an epifluorescence microscope (Zeiss AxioImager.M2) equipped with Zeiss "Plan-Apochromat" × 40/1.3 and × 63/1.40 oil objectives, a Coolsnap HQ$^2$ CCD camera (Photometrics) and Zeiss Axiovision software (version 4.2). Signals were quantified with ImageJ software (Rasband, W.S., ImageJ, US National Institutes of Health, Bethesda, Maryland, USA, http://rsb.info.nih.gov/ij/, 1997-2005). Overlapping profiles for two different antibodies on CFs were measured with the "RGB profiler" tool in ImageJ software. Images showing endogenous Aurora A staining were overexposed to reveal weak signals.

**Statistics**. All statistical analysis were performed with GraphPad Prism v6.05 (GraphPad Software). For the comparison of PSCS between cell lines, we performed two-tailed unpaired parametric $t$ tests or one-way analysis of variance (ANOVA) followed by Tukey's post hoc analysis, depending on the number of cell lines compared (two or more than two, respectively). When inhibitor was added to cell lines, we performed two-way ANOVA followed by Tukey's post hoc analysis for multiple comparisons. The numbers of experiments and of cells scored per experiment are indicated in the figure legends. Comparisons of fluorescence intensity from a representative experiment were analyzed in nonparametric two-tailed unpaired Mann–Whitney $U$-tests or Kruskal–Wallis tests followed by Dunn's multiple comparison test, depending on the number of cell lines compared (two or more, respectively). For all statistical tests, an alpha-risk of 0.05 was used.

**Data availability**. The detailed data (original pictures, detailed scoring, etc.) that support the findings of this study are available from the corresponding authors upon reasonable request.

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

## Acknowledgements

We thank Professor Jonathan Higgins for his pre-submission critical review of the manuscript, for the experiments he suggested and for the gift of reagents. We would like to thank Professor Yoshinori Watanabe (University of Tokyo, Japan), Dr Marie-Bérengère Troadec (UMR 6290 CNRS, France), and Dr Isabelle Bahon-Riedinger (Rennes University Hospital, France) for providing reagents. We thank Drs Erwan Watrin, Régis Giet, and Vincent Legagneux (IGDR, CNRS, Rennes, France) for stimulating and helpful discussions. G.F. was supported by a Ligue Contre le Cancer fellowship, G.E.-H. and L.M.-J. are investigators at the CNRS. S.M. and C.J. are investigators at INSERM. This work was funded by the French National Research Agency (ANR, project "EpiCentr"), the Région Bretagne (SAD grant), the Cancéropôle Grand Ouest, the Ligue Contre le Cancer (Comité Grand Ouest) and the Fondation ARC pour la Recherche sur le Cancer.

## Author contributions

G.E.-H. and L.M.-J. conducted experiments, analysed data, and wrote a first draft of the manuscript. G.F., F.-X.M. and S.M. conducted experiments. C.J. designed and supervised the project, analysed data, provided funding, and wrote the manuscript.

## Additional information

**Competing interests:** The authors declare no competing interests.

