## [Peer Review File · Nature Communications]

Reviewer #1 (Remarks to the Author):

This manuscript follows from work of Kunitoku et al. (2003) studying the phosphorylation of Cenp-A by Aurora A kinase. There are some interesting novel observations including the preferential localization of the p-Cenp-A to the inner centromere. The authors claim that Cenp-A phosphorylation by Aurora A protects chromosomes from cohesion fatigue. Unfortunately, there are fundamental flaws with interpretation that render it impossible to recommend publication of this work.

1. Kunitoku et al. showed that the phenotype of cells expressing Cenp-A S7A was a mitotic arrest with the majority of chromosomes at or near the metaphase plate and some chromosomes near the spindle poles. The extended arrest likely causes the chromosomes that are under bipolar tension at the metaphase plate to undergo cohesion fatigue. Thus the reason that Cenp-A S7A cells show higher levels of separated chromatids is likely not due to a direct effect of the phosphorylation in protecting cohesion but simply because cells expressing Cenp-A S7A are arrested in mitosis with many chromosomes under bipolar tension. The same result could explain the effects of Aurora A inhibition by siRNA or small molecule inhibitor (Fig 3). Consistent with this interpretation treatment with nocodazole greatly reduces chromatid separation of cells expressing Cenp-A S7A (Fig 5B).

2. The experiment that could test the authors interpretation would be to carefully synchronize cells expressing WT and S7A mutant Cenp-A so that they are arrested for the same length of time in a metaphase-like state and then compare the degree of chromatid separation. Figure 5C partially achieves this by treating populations with the proteasome inhibitor MG132. The text and figure indicate that chromatid separation is “higher” in the case of cells expressing S7A but neither the text nor the figure indicate that this difference reaches statistical significance. Moreover, the experiment is flawed since cells expressing S7A will already be arrested in mitosis prior to application of MG132, and they will have spent more time in mitosis and thus show more cohesion fatigue.

3. The authors indicate that depletion of Aurora B does not lead to chromatid separation and thus the cohesion protection is only attributable to Aurora A phosphorylation of Cenp-A. There are three issues. First, compromising Aurora B will disrupt kinetochore assembly and compromise spindle pulling forces necessary for cohesion fatigue. The authors allude to this possibility as an explanation which they cannot exclude. In reality, numerous studies have shown the importance of Aurora B in kinetochore assembly so this is almost a certainty. Second depending on the degree of depletion Aurora B is required for maintenance of the spindle assembly checkpoint particularly in cells with intact spindles. Thus, if Aurora B were strongly depleted, cells would be unable to maintain a mitotic arrest. Third Aurora B is reported to be important in the prophase pathway of cohesin removal. Therefore, depleting it may increase the amount of cohesin on chromosomes rendering them resistant to cohesion fatigue.

4. Figure 5G presents something of a contradiction that is unexplained by the authors model. Bub1 depletion leads to a decrease of Aurora A and p-Cenp-A but does not show chromatid separation above the control

5. From Figure 1D of exogenously expressed Cenp-A-HA, the authors conclude that only 3.7% percent of Cenp-A is phosphorylated at S7 based on the amount of the band with lower mobility. However, in Fig S2, anti-Cenp-A labeling of the endogenous protein in the top two panels show a band with reduced mobility in mitotic extracts present at nearly the same amount as the faster mobility band. If the slower band is endogenous p-Cenp-A, it undermines the authors rationale for why they are unable to detect unmodified Cenp-A with anti-Cenp-A antibody in the inner centromere.

Reviewer #2 (Remarks to the Author):

Eot-Houllier et.al. report a pathway that regulates centromeric cohesion in mitosis. They found that p-CENP-AS7 phosphorylated by both Aurora A and B mainly localizes to the inner centromere. Mild centromeric cohesion defects were observed in cells expressing the non-phosphorylatable S7A mutant or Aurora A-inhibited cells, but not Aurora B-inhibited cells. The cohesion defects can be ameliorated by treatment of nocodazole and worsened by treatment of MG132. Furthermore, they also found that Sgo1 localization at centromeres also decreased in cells expressing S7A, compared to cells expressing WT. Therefore, the authors claimed that Aurora A-dependent phosphorylation of CENP-A at the inner centromere acts as part of a final "safety switch" protecting bioriented chromosomes against cohesion fatigue. Finally, they also demonstrated that Bub1 functions at the top of this pathway by regulating Aurora A centromeric localization. Although p-CENP-AS7 phosphorylated by Aurora A and B has been known, it is interesting that this phosphorylation regulates centromeric cohesion in mitosis. These results in the report might be of interest to the field and broaden our understanding of cohesion regulation. As such, some concerns as follows still remain.

1. Depletion of either Aurora A or B significantly decreased the intensity of p-CENP-AS7 at centromeres to comparable levels, but only depletion of Aurora A, not Aurora B. induced cohesion defects. The authors also provided possible explanation for this observation in the discussion. However, it is also quite possible that other deficiency rather than loss of p-CENP-S7 in Aurora A-depleted cells leads to the observed cohesion defects. This possibility can be tested by examining if expression of CENP-AS7D or E (phosphorylation-mimetic mutants) can rescue the cohesion defects induced by Aurora A depletion or inhibition.

2. Bub1 depletion has been shown to relocate Sgo1 from centromeres to chromosome arms by decreasing p-H2A-T120, which significantly weakened centromeric cohesion. It is unclear whether loss of Aurora A and p-CENP-AS7 from centromeres by Bub1 depletion partially contributes to the Sgo1 relocalization and weakened centromeric cohesion. Again, to test this, Sgo1 localization and centromeric cohesion can be examined in Bub1-depleted cells expressing CENP-AS7D or E or with ectopically centromere-targeted Aurora A.

3. Although the authors have shown that Sgo1 centromeric intensity decreased in MG132-treated S7A cells, compared to WT cells. I am not convinced that decreased Sgo1 was attributed to the observed cohesion defects. To strengthen this, the authors can test if overexpression of Sgo1 could rescue the cohesion defects.

4. It has been well established that Sgo1 is recruited to centromeres by directly binding to p-H2A-T120, which mainly localizes at outer centromeres. This report shows that p-CENP-AS7 largely localizes at inner centromeres, and that no physical interactions between p-CENP-AS7 and Sgo1 were detected. Therefore, it is likely that p-CENP-AS7 contributes to Sgo1 localization to the centromere in an indirect manner.

Minor points:

1. Scale bars should be added into each staining figure.
2. In Figure 3C, p-CENP-AS7 should be quantified and WB blot results should also be shown.

Reviewer #3 (Remarks to the Author):

Centromeres in higher eukaryotes are specified by the CENP-A containing nucleosome. Phosphorylation of serine 7 on CENP-A was the first posttranslational modification of CENP-A identified. Despite this, several aspects of S7 phosphorylation are poorly understood. In this manuscript from Jaunlin and colleagues, the authors have uncovered a unique localization of S7 phosphorylated CENP-A to the inner centromere region. The authors propose that S7 phosphorylation has a role in protection of bi-oriented chromosomes against cohesion fatigue, a phenomenon of uncoordinated loss of sister chromatid cohesion as a result of sustained spindle tension applied to sister centromeres. The manuscript proposes that AuroraA and B both phosphorylate CENP-A, but only Aurora A loss contributes to precocious loss of sister chromatid cohesion, phenocopying a mutation of CENP-A S7. Finally, the authors suggest that CENP-A S7 phosphorylation is required for the maintenance of SGO1 when the centromere is under tension and this protects the chromatids from premature separation. Overall the identification of the inner centromere localization and the effect of the CENP-A S7A mutant are important contributions and well founded in the data. However, the mechanism by which CENP-A methylation is contributing to PSCS is not clear. See specific comments below that should be addressed prior to recommendation of the manuscript for publication.

Major points:

1. The localization of phosphorylated CENP-A to the inner centromere was surprising for two reasons. 1) it has not been observed before despite several papers on the subject and 2) phospho-antibody and bulk CENP-A antibody signal do not overlap. The authors used several different antibodies to show the inner centromere localization, and this rigor increases confidence that the localization is correct. The authors conclude that the amount of phosphorylated CENP-A is very low, and because the phospho-antibody is so good that it can detect the inner centromere pool, but the non-phospho antibodies do not due to lower affinity. However, the data for low levels of phosphorylated CENP-A are not as compelling as they need to be to explain the localization data. As it stands the data for low CENP-A is based on non-quantitative gel shift in Figure 1D. There is also concern that the assessment is based on the transfected CENP-A and not endogenous. With the quantitative assays in hand the authors should be able to estimate the absolute amount of phosphorylated and total CENP-A to determine how low the level of CENP-A phosphorylation really is. This is an important point and needs to be validated.
2. The IP in figure 4 needs to be better controlled. Considering the amount of inputs, the pulldowns are not very convincing, especially for aurora A where 0.5% input gives very prominent signal and IP is negligible. There needs to be a negative control shown in the IP for a chromatin associated protein that does not Co-IP for the blot to be meaningful. As a result the experiment is not convincing proof of the centromere associated Aurora A pool. The term "Ctrl. IP" is ambiguous.
3. Suppression of Aurora A and B both lead to loss of detectable CENP-A S7P, but only Aurora A causes PSCS. CENP-A phospho-mutants also lead to PSCS. This suggests that loss of CENP-A S7P is not sufficient to lead to PCSC, and Aurora A and the S7A mutant lead to PCSC via different mechanisms which does not fit the proposed hypothesis. Only a subset of cells show an increased PSCS, about 20% in each case. If Aurora A is suppressed when the S7 mutants are expressed (and endogenous CENP-A suppressed) does the PSCS phenotype get worse?
4. Based on the experiments presented in the study, the authors' conclusion that CENP-A S7 phosphorylation helps retention of SGO1 at centromeres is not very robust. In Figure 5A, B & C, no data is presented in control and CENP-A knocked down cells and only cell lines stably expressing CENP-A WT/S7A are used. If phosphorylation of CENP-A is important for the stabilization SGO1 at centromeres, CENP-A knockdown should mirror the effect of S7A mutant (completely or partially).
5. The data for Sgo1 loss in CENP-A S7A mutants only under tension (when treated with MG132) is descriptive, but the data in figure 5 do not provide a clear mechanism.
6. The graph in figure 5G is not a fair representation of the data. Figure 5B shows minimal reduction of Sgo1 in 75% of the cells in the S7A mutant, but the graph suggest that Sgo1 is down to 20%. This is just in the chromosomes that have undergone PSCS.

Minor Issues:

1. The authors claim that there is no difference in Bub1 in CENP-AWT vs CENP-A S7A mutant, but in the example shown in the bottom part of figure S4D the Bub1 levels certainly look different.
2. The pattern in 1Bc is unique. Is this one off or is this a consistent pattern?
3. Quantitation is needed for the pull fiber assays in 1C for the other 13 spreads that worked.

Response to Reviewers

**Aurora A-Dependent CENP-A Phosphorylation at Inner Centromeres Protects
Bioriented Chromosomes Against Cohesion Fatigue**

by

Grégory Eot-Houllier, Laura Magnaghi-Jaulin, Géraldine Fulcrand, François-Xavier
Moyroud, Solange Monier and Christian Jaulin

MS tracking number: NCOMMS-17-03693A

Original reviewers' comments are in blue.

Our responses are in black.

Changes to the manuscript text are in *italics*.

Reviewer #1:

This manuscript follows from work of Kunitoku et al. (2003) studying the phosphorylation of Cenp-A by Aurora A kinase. There are some interesting novel observations including the preferential localization of the p-Cenp-A to the inner centromere. The authors claim that Cenp-A phosphorylation by Aurora A protects chromosomes from cohesion fatigue. Unfortunately, there are fundamental flaws with interpretation that render it impossible to recommend publication of this work.

1. Kunitoku et al. showed that the phenotype of cells expressing Cenp-A S7A was a mitotic arrest with the majority of chromosomes at or near the metaphase plate and some chromosomes near the spindle poles. The extended arrest likely causes the chromosomes that are under bipolar tension at the metaphase plate to undergo cohesion fatigue. Thus the reason that Cenp-A S7A cells show higher levels of separated chromatids is likely not due to a direct effect of the phosphorylation in protecting cohesion but simply because cells expressing Cenp-A S7A are arrested in mitosis with many chromosomes under bipolar tension. The same result could explain the effects of Aurora A inhibition by siRNA or small molecule inhibitor (Fig 3). Consistent with this interpretation treatment with nocodazole greatly reduces chromatid separation of cells expressing Cenp-A S7A (Fig 5B).

Response to comment 1:

The reviewer suggested that the excess PSCS observed in CENP-AS7A cells was probably due to normal cohesion fatigue resulting from extended arrest in a metaphase-like state (due to the CENP-AS7A mutation), rather than a failure to protect cohesion against fatigue. However, the data presented in Figure 5B of our original manuscript show that treating CENP-AS7A cells with nocodazole for a short period (3 hours) leads to a PSCS rate similar to that of the wild-type control. Thus, sister chromatid separation occurred within the three hours preceding cell fixation in untreated CENP-AS7A cells. We can, therefore, assume that the PSCS observed in CENP-AS7A cells occurred within the

same three-hour period and did not result from accumulation over a longer period of time. Nevertheless, we acknowledge that our interpretation requires a stronger demonstration and we address this issue in our response to comment 2.

2. The experiment that could test the authors interpretation would be to carefully synchronize cells expressing WT and S7A mutant Cenp-A so that they are arrested for the same length of time in a metaphase-like state and then compare the degree of chromatid separation. Figure 5C partially achieves this by treating populations with the proteasome inhibitor MG132. The text and figure indicate that chromatid separation is “higher” in the case of cells expressing S7A but neither the text nor the figure indicate that this difference reaches statistical significance. Moreover, the experiment is flawed since cells expressing S7A will already be arrested in mitosis prior to application of MG132, and they will have spent more time in mitosis and thus show more cohesion fatigue.

Response to comment 2:

As requested by the reviewer, we synchronized cells at the G2/M boundary (double thymidine block followed by cdk1 inhibition). Cells were then released from synchronization and mitotic index and PSCS were scored at different time points. Figure 5E (new) shows that, at the time of release (time = 0) mitotic cells accounted for less than 3% of total cells. Thus, under our synchronization conditions, neither CENP-A-WT nor S7A cells accumulated in mitosis before RO-3306 block release. One hour after release, neither of these two cell types displayed PSCS as we score it (*i.e.* more than 50% of chromosomes displaying separation within the same cell), and an accumulation in mitosis was beginning to be detectable, this accumulation being more pronounced in S7A cells than in control cells. The rate of PSCS increased over time and was higher in S7A cells than in control cells. PSCS rates reached a 20% plateau at 3 h in S7A cells, were increased by the sustained tension induced by MG132 treatment and were abolished by nocodazole treatment.

These data can be interpreted in two ways:

1) In accordance with the reviewer's comment, S7A cells may be blocked in mitosis soon after RO-3306 release, consistent with the increase in mitotic index in these cells one hour after release, before the detection of PSCS. In this case, fatigue would, indeed, occur.

2) According to our model, S7A cells undergo normal mitosis, with chromosomes progressively losing cohesion due to an impairment of fatigue resistance. A loss of cohesion on one or a few chromosomes would lead to spindle assembly checkpoint activation and would be the cause, rather than the consequence, of S7A cell accumulation in mitosis. We ensured that our scoring was reliable by considering only cells in which the majority of chromosomes were separated as displaying PSCS. In this model, S7A cells harvested one hour after RO-3306 release, although not formally considered to display PSCS, would be expected to contain a larger number of unattached chromosomes than the control. However, it is technically difficult to score cells displaying a loss of cohesion for only one or a few chromosomes, because, in the absence of microtubule-disrupting drugs, chromosome spreads are more compact and individual CREST signals often overlap.

Nevertheless, synchronized wild-type control cells had PSCS rates of 20% following 5.5 hours of MG132 treatment, whereas this level was reached after only 2.5 hours of MG132 treatment in synchronized S7A cells (new Figure 5E). Thus, in our experimental system, cells expressing a wild-type CENP-A must be in a metaphase-like state (with chromosomes under bipolar tension) for 5.5 hours to reach PSCS levels of 20%, whereas it takes only 2.5 hours of MG132 treatment to reach the same PSCS level in S7A cells. No mitotic cells were detected at the time of RO-3306 block release (t_0). We can, therefore, assume that, at a given time point after the addition of MG132, both types of cells had spent identical times with their chromosomes under tension. However, both 3 h and 6 h after block release (2.5 and 5.5 hours, respectively, after the addition of MG132), the PSCS levels of S7A cells were, reproducibly, twice those of control cells. These results are not consistent with the excess PSCS observed in S7A cells being due to an extended mitotic block. By contrast, they are consistent with a role for CENP-AS7 phosphorylation in resistance to cohesion fatigue.

Moreover, the data presented in the new Figure 5C show that the phosphomimetic mutant CENP-AS7D is significantly more resistant to MG132-induced cohesion fatigue than the wild-type control. If the S7A mutation blocks cells in mitosis, leading to "normal" fatigue, it is difficult to see how a phosphomimetic mutant could lead to a reversion of this phenotype to levels below those of the control. The simplest interpretation is that CENP-AS7 phosphorylation (which is incomplete in wild-type CENP-A) contributes to resistance to cohesion fatigue.

Together, these results support our hypothesis that the mitotic arrest observed in CENP-AS7A-expressing cells is a consequence, rather than a cause, of accelerated cohesion fatigue.

Changes to the manuscript:

We have added the following to the results section:

" To ensure that the CENP-AS7A mutation do not cause an extended mitotic arrest in a metaphase-like stage that would eventually result in cohesion fatigue, and thus induce PSCS, cells were synchronized at the G2/M border, released for 30 minutes and supplemented with MG132. At the time of release, less than 2% mitotic cells were observed and we can, therefore, assume that, at a given time point after MG132 addition, both WT and S7A cells have spent identical amounts of time with their chromosomes under tension. However, both 3 h and 6 h after block release (2.5 and 5.5 hours, respectively, after MG132 addition), the PSCS levels in S7A cells were, reproducibly, twice those of control wild-type cells (Figure 5E, left graph). These results are not consistent with the excess PSCS observed in CENP-AS7A cells being due to an extended mitotic block. However, they are consistent with a role for CENP-AS7 phosphorylation in resistance to cohesion fatigue. The accumulation of untreated S7A cells in mitosis after block release (Figure 5E right graph) is, therefore, probably due to SAC activation in response to cohesion defects. Strikingly, cells expressing a phosphomimetic CENP-AS7D construct were more resistant to sustained tension than the wild-type control (Figure 5C) and, on the few separated sister chromatids observed, Sgo1 intensity had not decreased as markedly as in CENP-AS7A cells displaying PSCS (Figure 5D). We therefore conclude that p-CENP-

AS7 helps to protect against premature cohesion loss at centromeres when spindle force-induced tension is established during chromosome alignment."

The following has been added to the discussion:

"We report here that the expression of an unphosphorylatable mutant of CENP-AS7 leads to mitotic arrest and sister chromatid cohesion fatigue. It would be tempting to interpret the cohesion defects as a consequence of mutant-induced extended mitotic arrest, with chromosomes under tension. However, assays involving cell synchronization at the G2/M border showed that cells expressing a mutant CENP-AS7A construct were less resistant to sustained tension than CENP-AWT cells, although neither of these cell types was yet in mitosis after block release and both cell types had spent the same amount of time under tension following entry into mitosis (Figure 5E). Moreover, we found that a phosphomimetic mutant (CENP-AS7D) was significantly more resistant to cohesion fatigue than a wild-type control (Figure 5C). These data show that the mitotic arrest observed in CENP-AS7A expressing cells is a consequence (due to SAC activation) rather than a cause of greater cohesion fatigue. The accumulation of untreated S7A cells in mitosis after block release (Figure 5E right graph) is, therefore, probably due to SAC activation in response to cohesion defects. Strikingly, cells expressing a phosphomimetic CENP-AS7D construct were more resistant to sustained tension than the wild-type control (Figure 5C) and, on the few separated sister chromatids observed, Sgo1 intensity had no decreased to as great an extent as in CENP-AS7A cells displaying PSCS (Figure 5D). We therefore conclude that p-CENP-AS7 helps to protect against premature cohesion loss at centromeres when spindle force-induced tension is established during chromosome alignment."

Figures:

Figure 2, 3 and 5 have been completed with the results obtained with the CENP-AS7D mutant.

Figure 5E has been added.

3. The authors indicate that depletion of Aurora B does not lead to chromatid separation and thus the cohesion protection is only attributable to Aurora A phosphorylation of Cenp-A. There are three issues. First, compromising Aurora B will disrupt kinetochore assembly and compromise spindle pulling forces necessary for cohesion fatigue. The authors allude to this possibility as an explanation which they cannot exclude. In reality, numerous studies have shown the importance of Aurora B in kinetochore assembly so this is almost a certainty. Second depending on the degree of depletion Aurora B is required for maintenance of the spindle assembly checkpoint particularly in cells with intact spindles. Thus, if Aurora B were strongly depleted, cells would be unable to maintain a mitotic arrest. Third Aurora B is reported to be important in the prophase pathway of cohesin removal. Therefore, depleting it may increase the amount of cohesin on chromosomes rendering them resistant to cohesion fatigue.

Response to comment 3:

The reviewer commented that the role of Aurora B in cohesion protection, via the phosphorylation of CENP-AS7 and resistance to fatigue, may have been underestimated, because Aurora B is involved in the prophase pathway of cohesin removal, regulation of the spindle assembly checkpoint and the establishment of pulling forces between sister centromeres. Defects in these functions would strength cohesion, and possibly lead to premature exit from mitosis, with weakened opposing pulling forces, thereby resulting in an absence of PSCS detection in Aurora B-depleted S7A cells.

We agree with this comment and have modified the Discussion section accordingly.

Changes to the manuscript:

Results section, the results on the role of Aurora B localization have been moved to the supplemental information section (Figure S4).

Discussion section, the previous discussion on the role of Aurora B has been replaced with the following text:

"Both Aurora A and Aurora B phosphorylate CENP-A-S7 during mitosis, but only Aurora A depletion leads to sister chromatid cohesion defects. Aurora B-deficient cells do not present PSCS, despite the loss of a large proportion of CENP-AS7 phosphorylation. The function of CENP-AS7 phosphorylation by Aurora B therefore remains unclear. However, the depletion of Aurora B or the inhibition of its kinase activity leads to a loss of amphitelic attachment, preventing chromosome biorientation and tension establishment, and we show here that tension is required for CENP-AS7A-dependent PSCS. The possible role of Aurora B in protecting centromere cohesion against sustained tension at centromeres would, thus, be masked by its role in kinetochore/microtubule attachment processing. Moreover, Aurora B has been identified as involved in the "prophase pathway" of cohesin removal, and its inhibition or depletion would reinforce sister chromatid cohesion, making it impossible to assess its potential role in resistance to fatigue. "

Figures:

Figure S4 has been added.

4. Figure 5G presents something of a contradiction that is unexplained by the authors model. Bub1 depletion leads to a decrease of Aurora A and p-Cenp-A but does not show chromatid separation above the control

Response to comment 4:

The reviewer noted that, even though Bub1 acts upstream from the mechanisms protecting cohesion, its depletion does not lead to excess PSCS. As previously reported by Kitajima and coworkers (*Curr Biol* **15**, 353-359 (2005)), Bub1 depletion leads to the relocation of Shugoshin to chromosome arms and a weakening of centromeric cohesion. Chromosomes from cells lacking Bub1 typically display a greater intercentromere distance (1.3X) and stronger chromosome arm cohesion. We confirmed these results and checked for this specific phenotype in S7A and S7D (see Figure 6C). However, as the

chromosome arms were still attached in Bub1-depleted cells, these cells were not considered to display PSCS.

Changes to the manuscript:

Results section: a separate paragraph has been added to describe the results for Bub1. The following text has been added:

"Bub1 depletion leads to the loss of H2AT120 phosphorylation, the delocalization of Sgo1 from centromeres and a 1.3 times increase in inter-kinetochore distance that has been interpreted as a weakening of centromere cohesion. However, due to residual chromosome arm cohesion, Bub1-depleted cells do not display PSCS ^{41,42}. We investigated the phenotype of cells expressing mutant forms of CENP-AS7 lacking Bub1 (Figure 6C). As previously shown by Kitajima et al., inter-kinetochore distance increased significantly, by a factor of 1.25, upon Bub1 depletion in CENP-AWT cells (0.9 μm vs. 1.15 μm ; Figure 6C). S7A cells display no such plasticity, instead retaining their "open" conformation (inter-kinetochore distance of about 1.15 μm) regardless of Bub1 status. Conversely, cells expressing the CENP-AS7D phosphomimetic mutant adopt a "closed" conformation (inter-kinetochore distance of about 0.9 μm) in both the presence and absence of Bub1 depletion. Thus, inter-kinetochore distance seems to be under the control of CENP-AS7 phosphorylation. However, no Sgo1 signal was detected at centromeres in Bub1-depleted CENP-AS7D cells (Figure 6C, lower panels), confirming that p-CENP-AS7 is not a primary docking site for Sgo1, despite its role in retaining Sgo1 on separated sister chromatids."

Figures:

Figure 6C has been added.

5. From Figure 1D of exogenously expressed Cenp-A-HA, the authors conclude that only 3.7% percent of Cenp-A is phosphorylated at S7 based on the amount of the band with lower mobility. However, in Fig S2, anti-Cenp-A labeling of the endogenous protein in the top two panels show a band with

reduced mobility in mitotic extracts present at nearly the same amount as the faster mobility band. If the slower band is endogenous p-Cenp-A, it undermines the authors rationale for why they are unable to detect unmodified Cenp-A with anti-Cenp-A antibody in the inner centromere.

Response to comment 5:

The lower-mobility band (migrating between the bands corresponding to endogenous and exogenous CENP-A) seen in the mitotic extracts on Figure S2 (top panel, lanes 4 and 6) does not correspond to the endogenous form of CENP-A phosphorylated on serine 7, because it was detected only in cell lines expressing a CENP-A construct and not in untransfected cells. This band is often seen in mitotic extracts from cells expressing an exogenous form of CENP-A (with or without mutation), but is never detected in cells not expressing a CENP-A transgene. We attribute this band to a mitosis-specific degradation product of the exogenous CENP-A.

Changes to the manuscript:

None.

Reviewer #2:

Eot-Houllier et.al. report a pathway that regulates centromeric cohesion in mitosis. They found that p-CENP-AS7 phosphorylated by both Aurora A and B mainly localizes to the inner centromere. Mild centromeric cohesion defects were observed in cells expressing the non-phosphorylatable S7A mutant or Aurora A-inhibited cells, but not Aurora B-inhibited cells. The cohesion defects can be ameliorated by treatment of nocodazole and worsened by treatment of MG132. Furthermore, they also found that Sgo1 localization at centromeres also decreased in cells expressing S7A, compared to cells expressing WT. Therefore, the authors claimed that Aurora A-dependent phosphorylation of CENP-A at the inner centromere acts as part of a final "safety switch" protecting bioriented chromosomes against cohesion fatigue. Finally, they also demonstrated that Bub1 functions at the top of this pathway by regulating Aurora A centromeric localization. Although p-CENP-AS7 phosphorylated by Aurora A and B has been known, it is interesting that this phosphorylation regulates centromeric cohesion in mitosis. These results in the report might be of interest to the field and broaden our understanding of cohesion regulation. As such, some concerns as follows still remain.

1. Depletion of either Aurora A or B significantly decreased the intensity of p-CENP-AS7 at centromeres to comparable levels, but only depletion of Aurora A, not Aurora B. induced cohesion defects. The authors also provided possible explanation for this observation in the discussion. However, it is also quite possible that other deficiency rather than loss of p-CENP-S7 in Aurora A-depleted cells leads to the observed cohesion defects. This possibility can be tested by examining if expression of CENP-AS7D or E (phosphorylation-mimetic mutants) can rescue the cohesion defects induced by Aurora A depletion or inhibition.

Response to comment 1:

We agree that our conclusion that the Aurora A-mediated phosphorylation of CENP-AS7 is responsible for a loss of resistance to cohesion fatigue was based on a correlation (Aurora A can phosphorylate CENP-AS7, and both CENP-AS7 mutation and Aurora A inactivation lead to increased fatigue) rather than a demonstration. Stronger experimental evidence to support our conclusion was, therefore, required. As suggested by the reviewer, we constructed cell lines expressing phosphomimetic mutants of CENP-AS7 (CENP-AS7D and CENP-AS7E). As shown in the new Figure 2, cells expressing CENP-AS7D do not display PSCS. Moreover, these cells are more resistant to MG132-induced cohesion fatigue than the wild-type control (new Figure 5C), confirming a role for CENP-AS7 phosphorylation in fatigue resistance.

An investigation of the role of Aurora A showed that its inhibition in a CENP-AS7A context did not lead to significantly higher levels of PSCS than Aurora A inhibition or the expression of CENP-AS7A alone (new Figure 3C). This finding suggests that Aurora-A and CENP-AS7 act in the same functional compartment, protecting cohesion against fatigue. In addition, as suggested by the reviewer, we investigated whether the cohesion defects induced by Aurora-A inhibition could be rescued by the expression of CENP-AS7D. We found that CENP-AS7D expression completely abolished cohesion defects, even in the presence of Aurora A inhibition (Figure 3C). Cells expressing a CENP-AS7E mutant had the same phenotype (not shown). We therefore conclude that the Aurora A-mediated phosphorylation of CENP-AS7 protects against cohesion fatigue.

Changes to manuscript:

Results section, the following has been added:

"Cells expressing a phosphomimetic mutant (CENP-AS7D) did not display PSCS."

"A loss of p-CENP-AS7 associated with a loss of cohesion was also observed following the treatment of HeLa cells with a specific inhibitor of Aurora A (MLN8054; Figure 3B). Inhibitor efficiency was monitored by assessing the loss of the active form of Aurora A (p-Aurora AT288). As in the Aurora A depletion experiments, about 20% of the mitotic cells displayed premature sister chromatid separation, demonstrating a role for the kinase activity of Aurora A in protecting sister chromatid

cohesion. The depletion of Aurora-A and the inhibition of its kinase activity led to a loss of sister chromatid cohesion of a magnitude similar (~20%) to that observed in CENP-A-HA-S7A cells. Furthermore, the inhibition of Aurora A in CENP-AS7A cells did not aggravate the PSCS phenotype, and expression of the phosphomimetic mutant CENP-AS7D abolished the chromatid separation induced by Aurora-A inhibition (Figure 3C). Together, these results show that the Aurora A-driven phosphorylation of CENP-AS7 constitutes a novel mechanism for protecting sister chromatid cohesion at centromeres."

The following has been added to the discussion:

"By contrast, this study demonstrates the role of Aurora A in CENP-AS7 phosphorylation and the ensuring resistance to cohesion fatigue. Indeed, the depletion or inhibition of Aurora A phenocopies CENP-AS7A expression, and CENP-AS7D cells do not display PSCS, even upon Aurora A inhibition."

Figures:

Inhibition control has been added to Figure 3B.

Figure 3C has been added.

2. Bub1 depletion has been shown to relocate Sgo1 from centromeres to chromosome arms by decreasing p-H2A-T120, which significantly weakened centromeric cohesion. It is unclear whether loss of Aurora A and p-CENP-AS7 from centromeres by Bub1 depletion partially contributes to the Sgo1 relocalization and weakened centromeric cohesion. Again, to test this, Sgo1 localization and centromeric cohesion can be examined in Bub1-depleted cells expressing CENP-AS7D or E or with ectopically centromere-targeted Aurora A.

Response to comment 2:

Bub1 depletion leads to a loss of H2AT120 phosphorylation, the relocation of Sgo1 on the chromosome arms and a 1.3 time increase in inter-kinetochore distance during mitosis (Kitajima *et al.*

Curr Biol **15**, 353-359; 2005)). These observations have been interpreted as a weakening of centromeric cohesion upon Bub1 depletion. We investigated the phenotype of cells expressing mutant forms of CENP-AS7 with Bub1 depletion (new Figure 6). Sgo1 was lost from the centromere in Bub1-depleted wild-type control cells. We were able to reproduce the results obtained by Kitajima and coworkers for interkinetochore distances: these distances were significantly greater (1.25 times greater) following Bub1 depletion in WT cells (0.9 μm vs. 1.15 μm). Interestingly, S7A cells displayed no such plasticity, instead remaining in an "open" conformation (inter-kinetochore distance of about 1.15 μm) regardless of Bub1 status. Conversely, cells expressing the S7D phosphomimetic mutant adopt a "closed" conformation (inter-kinetochore distance of about 0.9 μm) in the presence and absence of Bub1 depletion. Thus, inter-kinetochore distance seems to be under the control of CENP-AS7 phosphorylation. However, no Sgo1 signal was recovered from the centromere in Bub1-depleted S7D cells.

These data suggest that the presence of Sgo1 at centromeres is not critical for the control of inter-kinetochore distance. However, there is a striking inverse correlation between inter-kinetochore distance and resistance to cohesion fatigue. Given our findings on the role of Bub1 in recruiting Aurora A to centromeres for CENP-AS7 phosphorylation, the most likely interpretation is that a mechanism involving p-CENP-AS7 accounts for resistance to fatigue when the chromosomes are subjected to tension and Sgo1 is relocated to the outer centromere.

Changes to manuscript:

A separate paragraph has been added to the results section, to describe the results for Bub1, and the following text has been added:

" Bub1 depletion leads to the loss of H2AT120 phosphorylation, the delocalization of Sgo1 from centromeres and a 1.3 times increase in inter-kinetochore distance that has been interpreted as a weakening of centromere cohesion. However, due to residual chromosome arm cohesion, Bub1-depleted cells do not display PSCS 41,42. We investigated the phenotype of cells expressing mutant forms of CENP-AS7 lacking Bub1 (Figure 6C). As previously shown by Kitajima et al., inter-

kinetochore distance increased significantly, by a factor of 1.25, upon Bub1 depletion in CENP-AWT cells (0.9 μm vs. 1.15 μm ; Figure 6C). S7A cells display no such plasticity, instead retaining their "open" conformation (inter-kinetochore distance of about 1.15 μm) regardless of Bub1 status. Conversely, cells expressing the CENP-AS7D phosphomimetic mutant adopt a "closed" conformation (inter-kinetochore distance of about 0.9 μm) in both the presence and absence of Bub1 depletion. Thus, inter-kinetochore distance seems to be under the control of CENP-AS7 phosphorylation. However, no Sgo1 signal was detected at centromeres in Bub1-depleted CENP-AS7D cells (Figure 6C, lower panels), confirming that p-CENP-AS7 is not a primary docking site for Sgo1, despite its role in retaining Sgo1 on separated sister chromatids."

The following has been added to the discussion:

"A role of p-CENP-A-S7 as a docking site for Sgo1 is an attractive hypothesis, but we observed no interaction in vitro between Sgo1 and a peptide corresponding to the sequence of the N-terminal part of CENP-A phosphorylated on serine 7 (not shown). Furthermore, we showed that CENP-AS7 mutation did not prevent Sgo1 recruitment, but weakened the anchoring of this protein to centromeres only when the sister chromatids were under tension. Finally, this hypothesis is not supported by the observation that expression of the phosphomimetic mutant CENP-AS7D in Bub-1-depleted cells does not rescue Sgo1 localization (Figure 6C)."

"Alternatively, the phosphorylation of CENP-AS7 may modify inner centromere chromatin plasticity, as suggested by the results obtained for chromosome spreads and stretched mitotic chromatin fibers, with Sgo1 stabilization at outer centromeres following tension establishment. This hypothesis is supported by the observation that inter-kinetochore distance is under the control of CENP-AS7 status (Figure 6C)."

Figures:

Figure 6 has been added.

3. Although the authors have shown that Sgo1 centromeric intensity decreased in MG132-treated S7A cells, compared to WT cells. I am not convinced that decreased Sgo1 was attributed to the observed cohesion defects. To strengthen this, the authors can test if overexpression of Sgo1 could rescue the cohesion defects.

Response to comment 3:

In our hands, the transient expression of Sgo1 led to massive cell death in less than 48 hours, whereas it took 80 hours to deplete endogenous CENP-A to levels allowing the observation of PSCS in S7A-expressing cells. We transfected cells with an Sgo1-FLAG expression vector 48 hours after the initial transfection with CENP-A siRNA, in an attempt to limit cell death, but the few Sgo1-positive cells we could detect were interphasic and systematically displayed intense uniform Sgo1-FLAG labeling very different from the punctuate, centromere-specific endogenous Sgo1 signal.

However, the decrease in the centromere Sgo1 signal in S7A cells displaying PSCS was not simply a consequence of cohesion loss, because Scc1-depleted cells had separated chromatids with a centromeric Sgo1 signal of an intensity similar to that in control cells (Figure S8).

Changes to manuscript:

We have added the following to the discussion:

"The prevention of CENP-AS7 phosphorylation leads to a loss of cohesion and the partial removal of Sgo1 from the centromeres of separated chromatids. It was not possible to demonstrate a causal relationship between these two observations, but cohesion defects are unlikely to be the sole cause of Sgo1 loss from centromeres because cohesion defects induced by cohesin depletion lead to the separation of sister chromatids bearing normal amounts of Sgo1 at their centromeres (Figure S8) ^{33,54}. Thus, Sgo1 loss from the centromeres of separated chromatids in S7A cells seems to be the consequence of both fatigue (because MG132-treated WT cells have significantly less Sgo1 on separated chromatids than on attached chromatids) and the S7A mutation, because the decrease in the

Sgo1 signal on both attached and unattached chromatids is more pronounced in S7A cells than in WT cells (Figure 5D)."

Figures:

Figure 5 has been modified to include S7D data.

Figure S8 has been added.

4. It has been well established that Sgo1 is recruited to centromeres by directly binding to p-H2A-T120, which mainly localizes at outer centromeres. This report shows that p-CENP-AS7 largely localizes at inner centromeres, and that no physical interactions between p-CENP-AS7 and Sgo1 were detected. Therefore, it is likely that p-CENP-AS7 contributes to Sgo1 localization to the centromere in an indirect manner.

Response to comment 4:

We agree with the reviewer's comment. Our results are, indeed, consistent with an indirect role of p-CENP-AS7 in the retention of Sgo1 retention at centromeres following the establishment of tension.

Changes to the manuscript:

In addition to the text added in response to comment 3, we now indicate in the discussion section that the role of p-CENP-AS7 in Sgo1 retention at centromeres must be indirect:

"We therefore favor a model in which p-CENP-AS7 indirectly stabilizes Sgo1 at centromeres after the establishment of tension, possibly by facilitating its binding to p-H2AT120, to protect against cohesion fatigue."

Minor points:

1. Scale bars should be added into each staining figure.

1) Scale bars have been added to each of the figures for staining.

2. In Figure 3C, p-CENP-AS7 should be quantified and WB blot results should also be shown.

2) The intensity of the p-CENP-AS7 signal has been quantified and the results have been added to Figure 3B. However, p-CENP-AS7 is not detectable on western blots of untreated unsynchronized cells (Figure S2, middle panel). We checked that Aurora A inhibition was effective, by western blotting for the active form of Aurora A (p-Aurora A T288) (Figure 3B).

We thank the reviewer for his/her suggestion to investigate the phenotype of cells expressing a CENP-AS7 phosphomimetic mutant, as this made it possible to reinforce and clarify several points in our study.

Reviewer #3:

Centromeres in higher eukaryotes are specified by the CENP-A containing nucleosome. Phosphorylation of serine 7 on CENP-A was the first posttranslational modification of CENP-A identified. Despite this, several aspects of S7 phosphorylation are poorly understood. In this manuscript from Jaunlin and colleagues, the authors have uncovered a unique localization of S7 phosphorylated CENP-A to the inner centromere region. The authors propose that S7 phosphorylation has a role in protection of bi-oriented chromosomes against cohesion fatigue, a phenomenon of uncoordinated loss of sister chromatid cohesion as a result of sustained spindle tension applied to sister centromeres. The manuscript proposes that AuroraA and B both phosphorylate CENP-A, but only Aurora A loss contributes to precocious loss of sister chromatid cohesion, phenocopying a mutation of CENP-A S7. Finally, the authors suggest that CENP-A S7 phosphorylation is required for the maintenance of SGO1 when the centromere is under tension and this protects the chromatids from premature separation. Overall the identification of the inner centromere localization and the effect of the CENP-A S7A mutant are important contributions and well founded in the data. However, the mechanism by which CENP-A methylation is contributing to PSCS is not clear. See specific comments below that should be addressed prior to recommendation of the manuscript for publication.

Major points:

1. The localization of phosphorylated CENP-A to the inner centromere was surprising for two reasons. 1) it has not been observed before despite several papers on the subject and 2) phospho-antibody and bulk CENP-A antibody signal do not overlap. The authors used several different antibodies to show the inner centromere localization, and this rigor increases confidence that the

localization is correct. The authors conclude that the amount of phosphorylated CENP-A is very low, and because the phospho-antibody is so good that it can detect the inner centromere pool, but the non-phospho antibodies do not due to lower affinity. However, the data for low levels of phosphorylated CENP-A are not as compelling as they need to be to explain the localization data. As it stands the data for low CENP-A is based on non-quantitative gel shift in Figure 1D. There is also concern that the assessment is based on the transfected CENP-A and not endogenous. With the quantitative assays in hand the authors should be able to estimate the absolute amount of phosphorylated and total CENP-A to determine how low the level of CENP-A phosphorylation really is. This is an important point and needs to be validated.

Response to comment 1:

We are aware that our method for determining the proportion of phosphorylated CENP-AS7 is not perfect. However, as signals from two different antibodies cannot be compared, it is unclear to which "quantitative assay" the reviewer is referring. It would have been more helpful to us had the kind of assay the reviewer has in mind been specified. Mass spectrometry techniques are not suitable for detecting CENP-AS7 phosphorylation, probably because the N-terminal CENP-A part of the molecule is very rich in positively charged amino acids. We were able to detect the presence of p-CENP-AS7 with specific antibodies, but not by MS (our unpublished observations). We tried to immunoprecipitate centromere chromatin and to normalize the CENP-A and p-CENP-AS7 signals against that for histone H4, but the H4 recovery results were erratic and uninterpretable, possibly due to variation of the length of the immunoprecipitated chromatin fragments.

We tried to quantify the proportion of p-CENP-AS7 to determine why we were unable to detect p-CENP-AS7 at inner centromeres with an anti-CENP-A antibody, but several other explanations may account for this lack of detection. For example, the anti-CENP-A antibody has a lower affinity for the phosphorylated form, as shown by competition experiments with synthetic peptides, and the antibody may be "titrated" on unphosphorylated CENP-A. Alternatively, other uncharacterized posttranslational

modifications close to serine 7 on p-CENP-AS7 may prevent epitope recognition (epitope masking) by the anti-CENP-A antibody.

Our data quantifying the shifted band present in wild-type mitotic cells but absent from mitotic CENP-AS7A cells is as quantitative as any western blot quantification can be. We agree that it does not provide an exact measurement, but it is accurate enough to identify p-CENP-AS7 as a small minority population relative to the unphosphorylated form. Moreover, as this report describes the role of Aurora A-mediated CENP-AS7 phosphorylation in resistance to cohesion fatigue, a precise measurement of the relative amounts of p-CENP-AS7 and the unphosphorylated form of this molecule does not appear to be essential. We have, therefore, removed this quantification from the manuscript and now indicate only that p-CENP-AS7 was present in much smaller quantities than unphosphorylated CENP-A.

Changes to the manuscript:

Results section:

The quantification has been removed and replaced with the following:

"The signal for the shifted band is much weaker than that for total CENP-A, indicating that p-CENP-AS7 accounts for only a small proportion of the CENP-A in the extracts."

2. The IP in figure 4 needs to be better controlled. Considering the amount of inputs, the pulldowns are not very convincing, especially for aurora A where 0.5% input gives very prominent signal and IP is negligible. There needs to be a negative control shown in the IP for a chromatin associated protein that does not Co-IP for the blot to be meaningful. As a result the experiment is not convincing proof of the centromere associated Aurora A pool. The term "Ctrl. IP" is ambiguous.

Response to comment 2:

The IP shown in Figure 2 confirms the immunofluorescence data. It is unsurprising that the amount of Aurora A recovered by centromere chromatin immunoprecipitation was small, because Aurora A is

localized at centrosomes, and was copurified during chromatin isolation (centrosomes are insoluble and contaminate chromatin preparations); it was therefore present in large amounts in the input. These IPs were eluted with a CENP-A peptide corresponding to the epitope recognized by the anti-CENP-A antibody. This strategy strongly limits the risks of false positives due to unspecific binding.

We were unable to identify a chromatin protein that was clearly not present at centromeres for the experiment suggested by the reviewer. However, we performed "reverse IP": we immunoprecipitated Aurora A from mitotic chromatin extracts and showed that we could co-immunoprecipitate CENP-A (new Figure 4C, right panel). Together with the immunofluorescence experiments, we believe that these data are robust enough to conclude that Aurora A is present at centromeres during mitosis.

Changes to the manuscript:

The following has been added to the results:

"In a reverse experiment, we immunoprecipitated Aurora A from mitotic chromatin extracts and demonstrated the co-immunoprecipitation of a fraction of NDC80 and CENP-A (Figure 4C, right panel)."

Figures:

The right panel has been added to Figure 4C.

3. Suppression of Aurora A and B both lead to loss of detectable CENP-A S7P, but only Aurora A causes PSCS. CENP-A phospho-mutants also lead to PSCS. This suggests that loss of CENP-A S7P is not sufficient to lead to PCSC, and Aurora A and the S7A mutant lead to PCSC via different mechanisms which does not fit the proposed hypothesis. Only a subset of cells show an increased PSCS, about 20% in each case. If Aurora A is suppressed when the S7 mutants are expressed (and endogenous CENP-A suppressed) does the PSCS phenotype get worse?

Response to comment 3:

We have performed the experiment suggested by the reviewer: scoring PSCS in S7A cells in which Aurora A is inhibited. We found that the PSCS phenotype was not significantly worse than that in cells expressing CENP-AS7A alone or with the inhibition of Aurora A only (new Figure 3C). The efficiency of inhibition was monitored by assessing the loss of Aurora AT288 phosphorylation upon inhibitor treatment (Figure 3B). This result suggests that Aurora-A and CENP-AS7 act in the same functional compartment to protect cohesion against fatigue. We also investigated whether the cohesion defects induced by Aurora-A inhibition could be rescued by the expression of a phosphomimetic mutant (CENP-AS7D). We found that CENP-AS7D expression completely abolished the cohesion defects, even when Aurora A was inhibited (Figure 3C). Cells expressing a CENP-AS7E mutant had the same phenotype (not shown). Remarkably, cells expressing CENP-AS7D were more resistant to MG132-induced cohesion fatigue than wild-type control cells (new Figure 5C). These data provide strong evidence of a role of Aurora-A-mediated CENP-AS7 phosphorylation in resistance to fatigue.

Changes to the manuscript:

The following has been added to the results:

"Inhibitor efficiency was monitored by assessing the loss of the active form of Aurora A (p-Aurora AT288). As in the Aurora A depletion experiments, about 20% of the mitotic cells displayed premature sister chromatid separation, demonstrating a role for the kinase activity of Aurora A in protecting sister chromatid cohesion. The depletion of Aurora-A and the inhibition of its kinase activity led to a loss of sister chromatid cohesion of a magnitude similar (~20%) to that observed in CENP-A-HA-S7A cells. Furthermore, the inhibition of Aurora A in CENP-AS7A cells did not aggravate the PSCS phenotype, and expression of the phosphomimetic mutant CENP-AS7D abolished the chromatid separation induced by Aurora-A inhibition (Figure 3C). Together, these results show that the Aurora A-driven phosphorylation of CENP-AS7 constitutes a novel mechanism for protecting sister chromatid cohesion at centromeres."

" Strikingly, cells expressing a phosphomimetic CENP-AS7D construct were more resistant to sustained tension than the wild-type control (Figure 5C) and, on the few separated sister chromatids

observed, Sgo1 intensity had not decreased as markedly as in CENP-AS7A cells displaying PSCS (Figure 5D)."

The following has been added to the discussion:

" By contrast, this study demonstrates the role of Aurora A in CENP-AS7 phosphorylation and the ensuring resistance to cohesion fatigue. Indeed, the depletion or inhibition of Aurora A phenocopies CENP-AS7A expression, and CENP-AS7D cells do not display PSCS, even upon Aurora A inhibition."

" Moreover, we found that a phosphomimetic mutant (CENP-AS7D) was significantly more resistant to cohesion fatigue than a wild-type control (Figure 5C). These data show that the mitotic arrest observed in CENP-AS7A expressing cells is a consequence (due to SAC activation) rather than a cause of greater cohesion fatigue."

"The combination of Aurora A inhibition with CENP-AS7A expression did not lead to more PSCS than Aurora A inhibition or CENP-AS7A expression alone. Thus, Aurora A and p-CENP-AS7 appear to be belong to the same epistasis group for PSCS phenotype. Moreover, expression of the phosphomimetic mutant CENP-AS7D rescues the cohesion defects induced by Aurora A inhibition. This, together with the role of CENP-AS7 as an Aurora A substrate in vitro (Kunitoku et al.) indicates that the most likely explanation to account for these observations is that Aurora A-mediated CENP-AS7 phosphorylation protects cohesion against fatigue."

Figures:

An inhibition control has been added to Figure 3B.

Figure 3C has been added.

Figures 2 and 5 have been modified to include S7D data.

4. Based on the experiments presented in the study, the authors' conclusion that CENP-A S7 phosphorylation helps retention of SGO1 at centromeres is not very robust. In Figure 5A, B & C, no

data is presented in control and CENP-A knocked down cells and only cell lines stably expressing CENP-A WT/S7A are used. If phosphorylation of CENP-A is important for the stabilization SGO1 at centromeres, CENP-A knockdown should mirror the effect of S7A mutant (completely or partially).

Response to comment 4:

We have checked that CENP-A depletion in HeLa cells (without the expression of CENP-A constructs) leads to weaker resistance to fatigue. We show that untreated HeLa cells display 10% PSCS following CENP-A depletion. This phenotype is worsened by sustained tension (MG132 treatment) and almost abolished by treatment with nocodazole (Figure S6). We measured Sgo1 signal intensity: Sgo1 signals were much weaker on separated sister chromatids upon CENP-A depletion. Moreover, Sgo1 intensity was also significantly lower (albeit to a lesser extent) on joined sister chromatids upon CENP-A depletion. In addition, mock depletion (control) in MG132-treated HeLa cells showed that Sgo1 signal intensity was slightly weaker on separated sister chromatids than on intact chromosomes. This last finding might suggest that loss of cohesion destabilizes Sgo1. However, this phenomenon seems to be specific to fatigue-induced cohesion loss, because Scc1-depleted cells have separated chromatids with a centromeric Sgo1 signal of an intensity similar to that of the control (Figure S8).

Taken together with our previous results for WT/S7A/S7D expressing cells, these data indicate that p-CENP-AS7 plays a role in stabilizing Sgo1 at centromeres when sister chromatids are subjected to tension.

Changes to the manuscript:

We have added the following to the results:

"CENP-A depletion in HeLa cells mimics the expression of CENP-AS7A in terms of both weakened resistance to tension and the lack of Sgo1 stabilization on fatigue-induced separated chromatids, confirming the role of CENP-A in these two functions (Figure S6A-C)"

The following has been added to the discussion:

" The prevention of CENP-AS7 phosphorylation leads to a loss of cohesion and the partial removal of Sgo1 from the centromeres of separated chromatids. It was not possible to demonstrate a causal relationship between these two observations, but cohesion defects are unlikely to be the sole cause of Sgo1 loss from centromeres because cohesion defects induced by cohesin depletion lead to the separation of sister chromatids bearing normal amounts of Sgo1 at their centromeres (Figure S8) 33,54. Thus, Sgo1 loss from the centromeres of separated chromatids in S7A cells seems to be the consequence of both fatigue (because MG132-treated WT cells have significantly less Sgo1 on separated chromatids than on attached chromatids) and the S7A mutation, because the decrease in the Sgo1 signal on both attached and unattached chromatids is more pronounced in S7A cells than in WT cells (Figure 5D)."

Figures:

Figure 5 has been modified to include S7D data.

Figure S6 has been added.

Figure S8 has been added.

5. The data for Sgo1 loss in CENP-A S7A mutants only under tension (when treated with MG132) is descriptive, but the data in figure 5 do not provide a clear mechanism.

Response to comment 5:

Our data show that the Aurora A-mediated phosphorylation of CENP-AS7 is involved in stabilizing Sgo1 at centromeres and in resistance to cohesion fatigue. We tried to overexpress an Sgo1-FLAG construct, to assess the role of Sgo1 in resistance to fatigue, but the few Sgo1-FLAG-positive cells we were able to detect were all interphasic and systematically displayed intense uniform Sgo1-FLAG labeling unlike the punctate centromere-specific endogenous Sgo1 signal. Thus, although perfectly logical, we cannot as yet link, with any degree of certainty, the decrease in Sgo1 recruitment at

centromeres to lower resistance to fatigue in S7A cells or CENP-A-depleted HeLa cells. This point is now discussed in the "discussion" section.

We report here the first identification of a cellular process involved in regulating resistance to cohesion fatigue. The detailed molecular mechanisms involved in this regulation remain to be determined and are currently under investigation in our laboratory.

Changes to the manuscript:

The following has been added to the discussion (same change as for our answer to comment 4):

" The prevention of CENP-AS7 phosphorylation leads to a loss of cohesion and the partial removal of Sgo1 from the centromeres of separated chromatids. It was not possible to demonstrate a causal relationship between these two observations, but cohesion defects are unlikely to be the sole cause of Sgo1 loss from centromeres because cohesion defects induced by cohesin depletion lead to the separation of sister chromatids bearing normal amounts of Sgo1 at their centromeres (Figure S8) 33,54. Thus, Sgo1 loss from the centromeres of separated chromatids in S7A cells seems to be the consequence of both fatigue (because MG132-treated WT cells have significantly less Sgo1 on separated chromatids than on attached chromatids) and the S7A mutation, because the decrease in the Sgo1 signal on both attached and unattached chromatids is more pronounced in S7A cells than in WT cells (Figure 5D)."

Figures:

Figure 5 has been modified to include S7D data.

Figure S8 has been added.

6. The graph in figure 5G is not a fair representation of the data. Figure 5B shows minimal reduction of Sgo1 in 75% of the cells in the S7A mutant, but the graph suggest that Sgo1 is down to 20%. This is just in the chromosomes that have undergone PSCS.

Response to comment 6:

We fully agree with the reviewer and apologize for this misleading mistake. Figure 5G has been removed from the revised version of our manuscript, and the corresponding data are now described and discussed in the text.

Minor Issues:

1. The authors claim that there is no difference in Bub1 in CENP-AWT vs CENP-A S7A mutant, but in the example shown in the bottom part of figure S4D the Bub1 levels certainly look different.

1) We agree that the image in Figure S4D does not properly illustrate the Bub1 intensity measurements shown on the corresponding graph. This image has now been replaced with a more representative image (Figure S7A, lower panels).

2. The pattern in 1Bc is unique. Is this one off or is this a consistent pattern?

2) The pattern shown in figure 1Bc is not unique and was observed in 2% of all the chromosomes scored. We have reduced the complexity of the figure by not indicating the proportion of each of the CENP-A pattern subcategories.

3. Quantitation is needed for the pull fiber assays in 1C for the other 13 spreads that worked.

3) The other 13 stretched chromatin fibers from mitotic centromeres that we obtained are now displayed in Figure S1E.

Reviewer #1 (Remarks to the Author):

The authors have done a credible job responding the reviews. The addition of the phosphomimetic data particularly strengthens the argument that the phosphorylation of CenpA plays a role in interchromatid cohesion. And they are to be commended for carrying out the synchronization studies to better clarify the effects on cohesion fatigue. Some remaining areas of clarification regarding my original suggestions.

1. Kunitoku et al. reported chromosome alignment defects. This is quite distinct from cohesion fatigue, which is not an alignment failure but rather a separation of chromatids. It was not clear in the original ms whether the authors believe the original observations of Kunitoku et al. of chromosome alignment defects are in error, a misinterpretation of cohesion fatigue. They appear to make the claim that initial fatigue of a few chromosomes activates the spindle assembly checkpoint allowing cells to arrest. These arrested cells then undergo further cohesion fatigue. While I am sympathetic to the notion that detecting low levels of cohesion fatigue in cells expressing S7A is difficult in chromosome spreads, the early separating chromosomes should show checkpoint markers at their kinetochores, which is noted in the Kunitoku et al paper. In short the authors should clarify in the ms, not just in the rebuttal, whether they see alignment defects, cohesion fatigue or both in the S7A expressing cells as the cause of mitotic delay.

2. In truth the authors' rebuttal to my original comment 1 was not entirely satisfactory. The treatment in nocodazole may have allowed reestablishment of cohesin linkages as has been reported in budding yeast (Ocampo-Hafalla et al., 2007, Chromosoma 116:531) or through DNA catenation. However, their carrying out of the synchronization experiments in response to my comment 2 addressed the major issue.

Reviewer #2 (Remarks to the Author):

Thanks for the authors' efforts to revise and improve the manuscript. I am satisfied with the revisions and recommend for the publication in NC.

Reviewer #3 (Remarks to the Author):

Overall I think the manuscript provides interesting and novel insight into CENP-A phosphorylation at serine 7. Several of the issues I raised in the original review have been well addressed in this revised manuscript. The new data provided in figure 3D showing only modest difference in the degree of PSCS between CENP-A S7A expressing cells and those also treated with Aurora A inhibitor supports the idea that these two perturbation are functioning via the same mechanism to induce loss of cohesion. The fact that CENP-A S7D is able to suppress the AurkA inhibition phenotype is also very compelling. However, the data in figure 3A are still confusing to me. How is it that both Aurora B and A lead to S-phospho loss, but only Aurora A leads to increased PSCS. This suggest that phosphorylation is dispensable, and it is all about AurkB. I cannot reconcile these data with the S7D data cited above.

The new IP data showing IP of aurora A results in co-purification of CENP-A is good; however, it still suffers from lack of a good negative control. The direct association between CENP-A and Aurora A was shown by Kunitoko et al. 2003 in a very convincing and complete analysis. I would suggest that panel 4C be removed from the manuscript and that Kunitoko paper referened.

The data in Figure 6C are interesting, but the way the figures are cropped is very odd, entire spreads show be shown.

Reviewer #1 (Remarks to the Author):

The authors have done a credible job responding the reviews. The addition of the phosphomimetic data particularly strengthens the argument that the phosphorylation of CenpA plays a role in interchromatid cohesion. And they are to be commended for carrying out the synchronization studies to better clarify the effects on cohesion fatigue. Some remaining areas of clarification regarding my original suggestions.

1. Kunitoku et al. reported chromosome alignment defects. This is quite distinct from cohesion fatigue, which is not an alignment failure but rather a separation of chromatids. It was not clear in the original ms whether the authors believe the original observations of Kunitoku et al. of chromosome alignment defects are in error, a misinterpretation of cohesion fatigue. They appear to make the claim that initial fatigue of a few chromosomes activates the spindle assembly checkpoint allowing cells to arrest. These arrested cells then undergo further cohesion fatigue. While I am sympathetic to the notion that detecting low levels of cohesion fatigue in cells expressing S7A is difficult in chromosome spreads, the early separating chromosomes should show checkpoint markers at their kinetochores, which is noted in the Kunitoku et al paper. In short the authors should clarify in the ms, not just in the rebuttal, whether they see alignment defects, cohesion fatigue or both in the S7A expressing cells as the cause of mitotic delay

Response to comment 1:

Defective chromosome alignment is characterized by the failure to align chromosomes along metaphase plate. Kunitoku and collaborators (2003) reported such an alignment defect in CENP-AS7A-expressing cells and suggested that this defect would result from improper microtubule attachment to kinetochores. This interpretation was based on the fact that misaligned chromosomes were found enriched for checkpoint markers at their kinetochores. We also observed chromosome misalignment in our S7A cell lines (see figure below).

WT

S7A

It is now well documented that unscheduled sister chromatid separation also causes chromosome alignment defects. Moreover, cohesion fatigue also leads to misaligned chromosomes displaying strong SAC markers labeling at their kinetochores (Stevens et al, 2011, *PLoS ONE* **6**, e22969), precluding to discriminate between cohesion fatigue and defective microtubule/KT attachment on this sole basis. Therefore, given our observation on cohesion fatigue, the simplest interpretation is that, in *S7A*-expressing cells, alignment defects would be the consequence of cohesion loss rather than of a failure to connect microtubules to kinetochores.

Changes to the manuscript:

The following text has been added to the "Discussion" section:

" It has been reported that overexpression of CENP-AS7A leads to chromosome alignment defects and the authors suggested that this defect would result from improper microtubule attachment to kinetochores¹³. However, chromosome misalignment is a known phenotype associated with unscheduled sister chromatid separation ^{36,37}. Therefore, given our observations on cohesion fatigue, the simplest interpretation is that, in CENP-AS7A-expressing cells, alignment defects would be the consequence of cohesion loss rather than of a failure to connect microtubules to kinetochores."

2. In truth the authors' rebuttal to my original comment 1 was not entirely satisfactory. The treatment in nocodazole may have allowed reestablishment of cohesin linkages as has been reported in budding yeast (Ocampo-Hafalla et al., 2007, *Chromosoma* 116:531) or through DNA catenation. However, their carrying out of the synchronization experiments in response to my comment 2 addressed the major issue.

Response to comment 2:

We acknowledge that our initial analysis was not sufficient to conclude that the S7A mutation induces enhanced cohesion fatigue and we thank the reviewer for his/her constructive comments and suggestions.

Reviewer #2 (Remarks to the Author):

Thanks for the authors' efforts to revise and improve the manuscript. I am satisfied with the revisions and recommend for the publication in NC.

Response to comment:

We are grateful to the reviewer for his/her recommendation.

Reviewer #3 (Remarks to the Author):

Overall I think the manuscript provides interesting and novel insight into CENP-A phosphorylation at serine 7. Several of the issues I raised in the original review have been well addressed in this revised manuscript. The new data provided in figure 3D showing only modest difference in the degree of PSCS between CENP-A S7A expressing cells and those also treated with Aurora A inhibitor supports the idea that these two perturbation are functioning via the same mechanism to induce loss of cohesion. The fact that CENP-A S7D is able to suppress the Aurka inhibition phenotype is also very compelling. However, the data in figure 3A are still confusing to me. How is it that both Aurora B and A lead to S-phospho loss, but only Aurora A leads to increased PSCS. This suggest that phosphorylation is dispensable, and it is all about AurkB. I cannot reconcile these data with the S7D data cited above.

Response to comment:

We agree that the role of Aurora B is puzzling. However, as another reviewer pointed out, a possible function for Aurora B in participating to CENP-AS7 phosphorylation and resistance to cohesion fatigue cannot be assessed since Aurora B is involved in the prophase pathway of cohesin removal, regulation of the spindle assembly checkpoint and the establishment of pulling forces between sister centromeres. Defects in these functions would strength cohesion, and possibly lead to premature exit from mitosis, with weakened opposing pulling forces, thereby resulting in an absence of PSCS detection in Aurora B-depleted S7A cells.

This question was addressed in the previous revised version of our manuscript and the discussion on the role of Aurora B was replaced with the following text:

"Both Aurora A and Aurora B phosphorylate CENP-A-S7 during mitosis, but only Aurora A depletion leads to sister chromatid cohesion defects. Aurora B-deficient cells do not present PSCS, despite the loss of a large proportion of CENP-AS7 phosphorylation. The function of CENP-AS7 phosphorylation by Aurora B therefore remains unclear. However, the depletion of Aurora B or the inhibition of its kinase activity leads to a loss of amphitelic attachment, preventing chromosome biorientation and tension establishment, and we show here that tension is required for CENP-AS7A-dependent PSCS. The possible role of Aurora B in protecting centromere cohesion against sustained tension at centromeres would, thus, be masked by its role in kinetochore/microtubule attachment processing. Moreover, Aurora B has been identified as involved in the "prophase pathway" of cohesin removal,

and its inhibition or depletion would reinforce sister chromatid cohesion, making it impossible to assess its potential role in resistance to fatigue. "

The new IP data showing IP of aurora A results in co-purification of CENP-A is good; however, it still suffers from lack of a good negative control. The direct association between CENP-A and Aurora A was shown by Kunitoko et al. 2003 in a very convincing and complete analysis. I would suggest that panel 4C be removed from the manuscript and that Kunitoko paper referenced.

Response to comment:

We are aware that these IPs lack a good negative control but, as we mentioned earlier, we could not identify a chromatin protein that was clearly not present at centromere (all histones are present at centromeres and even telomere proteins such as TRF2 have been detected at centromeres). The goal of these experiments was to confirm, using an independent approach, our immunofluorescence data showing that Aurora A is present at centromeres. Kunitoku and collaborators have shown a direct interaction between recombinant tagged Aurora A and CENP-A which is distinct from our observation that endogenous Aurora A and CENP-A co-localize at centromeres. Thus, we believe that our observations are more complementary than redundant with theirs. However, we agree that our IP data do not represent a key finding in our report and panel 4C has been removed from the main manuscript and inserted into the "supplemental data" section (new Figure S4E). Kunitoku's paper was referenced in the "results" section of our manuscript as follows: "*These results demonstrate the presence of endogenous Aurora A at centromeres and are consistent with previous results showing interactions between recombinant CENP-A and Aurora A*¹³"

The data in Figure 6C are interesting, but the way the figures are cropped is very odd, entire spreads show be shown.

Response to comment:

We had tried to illustrate both the lack of Sgo1 recruitment following Bub1 depletion and the differences in inter-kinetochores distances. We realize that it can be confusing and the pictures in Figure 6C have been replaced with entire spreads, as suggested by the reviewer.